# TRIDENT: Tri-Modal Molecular Representation Learning with Taxonomic Annotations and Local Correspondence

**Feng Jiang**[1]    **Mangal Prakash**[2]    **Hehuan Ma**[1]    **Jianyuan Deng**[2]    **Yuzhi Guo**[1]
**Amina Mollaysa**[2]    **Tommaso Mansi**[2]    **Rui Liao**[2]    **Junzhou Huang**[1]

[1]University of Texas at Arlington
[2]Johnson & Johnson Innovative Medicine
{fxj8843, hehuan.ma, yuzhi.guo}@mavs.uta.edu    {jzhuang}@uta.edu
{MPraka12, JDeng34, MAminanm, TMansi, RLiao2}@ITS.JNJ.com

## Abstract

Molecular property prediction aims to learn representations that map chemical structures to functional properties. While multimodal learning has emerged as a powerful paradigm to learn molecular representations, prior works have largely overlooked textual and taxonomic information of molecules for representation learning. We introduce TRIDENT, a novel framework that integrates molecular SMILES, textual descriptions, and taxonomic functional annotations to learn rich molecular representations. To achieve this, we curate a comprehensive dataset of molecule-text pairs with structured, multi-level functional annotations. Instead of relying on conventional contrastive loss, TRIDENT employs a volume-based alignment objective to jointly align tri-modal features at the global level, enabling soft, geometry-aware alignment across modalities. Additionally, TRIDENT introduces a novel local alignment objective that captures detailed relationships between molecular substructures and their corresponding sub-textual descriptions. A momentum-based mechanism dynamically balances global and local alignment, enabling the model to learn both broad functional semantics and fine-grained structure-function mappings. TRIDENT achieves state-of-the-art performance on 18 downstream tasks, demonstrating the value of combining SMILES, textual, and taxonomic functional annotations for molecular property prediction. Our code and data are available at `https://github.com/uta-smile/TRIDENT`.

## 1 Introduction

Molecular representation learning, which converts complex chemical structures into computational features, has been instrumental in advancing various aspects of drug discovery including virtual screening, and molecular design [15, 4, 34]. Multi-modal molecular models further enhance representation quality by integrating structural, textual, and functional information, enabling better generalization and predictive performance [16]. These approaches hold promise for unlocking deeper insights into chemical space and accelerating the discovery of therapeutic compounds with desired properties.

However, current multimodal approaches [21, 35] face three key limitations: **(1) Overlooking fine-grained annotations across taxonomies**: Most existing methods simplify the representation of molecules by focusing on unified functional descriptions, neglecting the nuanced annotations provided by different taxonomic systems. As illustrated in Figure 2, the same molecule may have distinct emphases depending on the taxonomy: for example, the LOTUS Tree [33] taxonomy highlights

39th Conference on Neural Information Processing Systems (NeurIPS 2025).

natural product classifications, whereas the MeSH (Medical Subject Headings) Tree [14] taxonomy emphasizes medical functionalities of the same molecule. Ignoring these taxonomy-specific, fine-grained annotations risks reducing molecules to flat entities, thereby failing to capture the multi-faceted and structured nature of chemical functions. **(2) Alignment limitations**: Aligning modalities such as molecular structures, textual descriptions, and taxonomic functional annotations is inherently complex. Existing methods rely on pairwise alignment schemes anchored to a single modality, which struggle to model the interdependencies across all modalities [43, 17, 38, 3], particularly when one modality encodes nested or multi-level information [5]. **(3) Neglect of local correspondences**: Many approaches focus exclusively on molecule-level alignment, disregarding the fine-grained relationships between molecular substructures (e.g., functional groups) and their corresponding sub-textual descriptions. This omission limits the expressivity of the learned representations and constrains their applicability in molecular property prediction tasks.

To address these limitations, we introduce the TRIDENT (Tri-modal Representation Integrating Descriptions, Entities, and Taxonomies) framework for molecules that jointly models molecular SMILES, textual descriptions, and multi-faceted Hierarchical Taxonomic Annotation (HTA). Central to TRIDENT is the HTA modality, which organizes molecular function across hierarchical classification levels. We curate a high quality dataset of 47,269 *<SMILES, Text, HTA>* triplets from PubChem [13], annotated under 32 classification systems. To tackle the challenge of aligning these diverse modalities, TRIDENT leverages a volume-based contrastive loss, enabling soft, geometry-aware alignment of all three modalities. While recently proposed for general-purpose modality alignment [5], we extend this formulation to the molecular domain for the first time, where the modalities are structurally diverse and include taxonomic semantic labels. Furthermore, TRIDENT introduces a novel local alignment module that links molecular substructures to their associated sub-textual descriptions, capturing fine-grained structure–function relationships. A momentum-based balancing mechanism dynamically integrates global and local alignments to optimize the representation learning process (see Figure 1 for an overview).

We demonstrate that TRIDENT achieves consistent and substantial improvements over existing molecular representation learning methods. Our framework sets a new benchmark, delivering state-of-the-art performance across 11 downstream molecular property prediction tasks on established benchmarks, while remaining modular and flexible, allowing integration of different modality encoders without the need for architectural modifications. We have also created a high-quality, comprehensive dataset of molecule-text-function triplets, which forms the foundation for this work and future research. To summarize, we make the following contributions:

- Introducing a Hierarchical Taxonomic Annotation (HTA) modality for molecules, supported by a newly curated high-quality multimodal dataset consisting of 47,269 *<SMILES, Text, HTA>* triplets annotated across 32 diverse taxonomic classification systems. This enables a structured, multi-level functional understanding of molecules, providing a novel resource for molecular representation learning.

- A unified global–local alignment strategy that integrates a volume-based contrastive loss for tri-modal global alignment with a novel local alignment module for substructure–subtext correspondence, dynamically balanced via a momentum-based mechanism.

- Demonstrated state-of-the-art performance across 11 molecular property prediction tasks, validating the effectiveness of hierarchical taxonomic annotations as a modality, the proposed alignment strategies, and the quality of the curated dataset.

## 2 Related Works

### 2.1 Molecule-Text Multimodal Learning

Recent advancements in molecular representation learning have demonstrated the power of multi-modal approaches that integrate information from molecular graphs, SMILES strings, and textual descriptions to enhance property prediction and drug discovery. Graph Neural Networks (GNNs) have become the backbone of graph-based methods, with models like GROVER [30] and MolCLR [36] leveraging contrastive learning to produce richer molecular embeddings. Multimodal models such as KV-PLM [39] and MolT5 [7] treat SMILES and text as separate languages for pre-training via auto-encoding objectives, while MoMu [35] and MoleculeSTM [16] utilize independent encoders

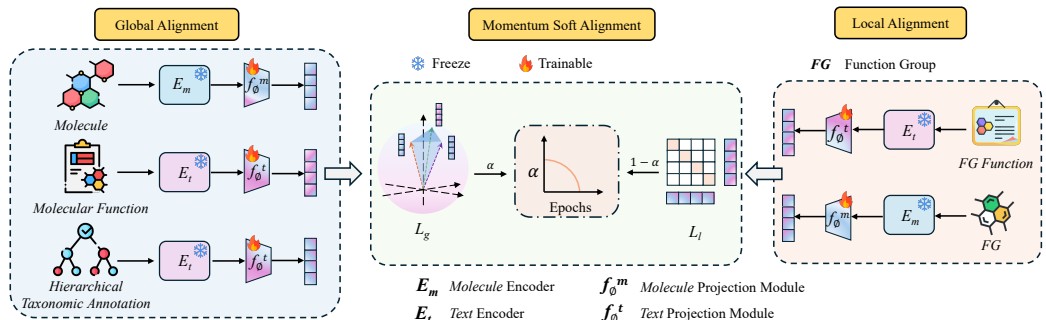

Figure 1: **Overview of TRIDENT.** TRIDENT jointly models molecular SMILES, natural language descriptions, and Hierarchical Taxonomic Annotations (HTAs) to learn rich molecular representations. The framework employs a volume-based contrastive loss for soft global tri-modal alignment and a local alignment module that links molecular substructures to sub-text spans. A momentum-based mechanism dynamically balances the contribution of global and local objectives during training. This multimodal, multi-level alignment enables precise and semantically grounded molecular understanding.

with cross-modal contrastive learning to align graphs and texts. MolFM [21] extends this paradigm by incorporating molecular structures, biomedical texts, and knowledge graphs to capture more comprehensive molecular relationships. However, despite this progress, the textual modality in existing models often derives from unstructured or single-layered descriptions, limiting the capacity to represent molecular functions across diverse biological roles and hierarchical categories. This lack of structured semantic alignment limits the ability of models to reason over complex molecular behaviors and relationships. Our work addresses this gap by introducing a high quality dataset to incorporate hierarchical taxonomic annotations for molecules, learning fine-grained hierarchical molecule-function relationships.

## 2.2 Contrastive Learning for Multimodal Alignment

Contrastive learning has emerged as a powerful strategy for aligning representations across modalities. Seminal models such as CLIP [28] demonstrated effective image-text alignment, inspiring extensions to other domains including audio (CLAP) [8], video (CLIP4Clip) [20], and point clouds (Point-CLIP) [40]. These models typically learn by pulling semantically similar cross-modal pairs closer while pushing dissimilar ones apart. More recent approaches such as CLIP4VLA [32], ImageBind [9], and LanguageBind [43] explore multimodal fusion, often anchoring learning around a central modality like images or text. GRAM [5] advances this direction by introducing geometry-aware volume based contrastive objective, but it primarily focuses on audio-video-text pairs without structured semantic hierarchies. In bioinformatics, multimodal learning has shown promise in integrating diverse biological data sources [11, 6, 24], such as molecular structures, protein sequences, and biomedical text, to enhance understanding of complex biological systems and accelerate drug discovery [12, 22, 23]. Unlike existing methods, our TRIDENT framework tackles the unique challenges of molecule-text alignment by incorporating hierarchical taxonomic relationships to capture functional semantics, and introducing global and local alignment modules with momentum-based mechanism. This enables fine-grained substructure-function correspondence and a richer multimodal embedding space tailored to molecular understanding.

## 3 Method

In this section, we provide a detailed introduction to the implementation of the TRIDENT framework, as illustrated in Figure 1 which addresses the shortcomings of existing methods in capturing a structured understanding of molecular functions across different hierarchical functional categories.

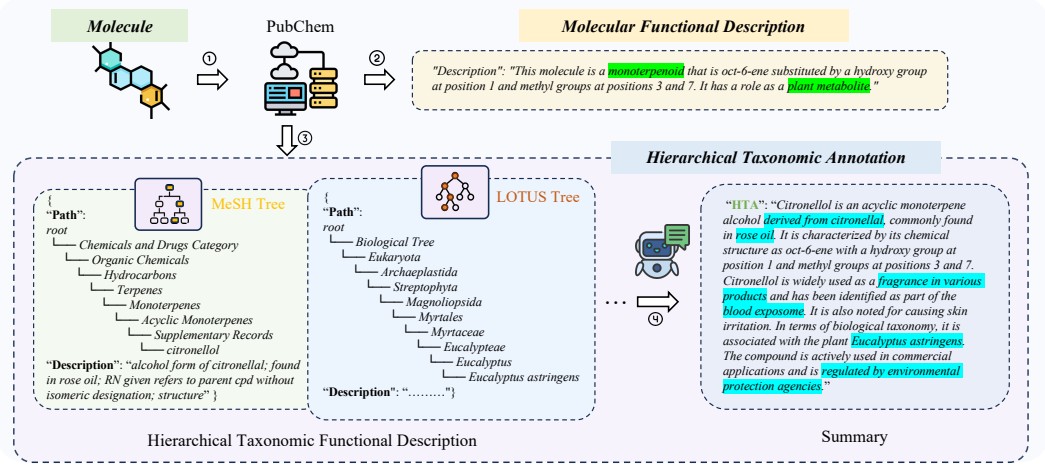

Figure 2: Traditional molecular functional descriptions are typically obtained by inputting a molecule into PubChem, where a general functional annotation is provided, as shown in Steps 1 and 2 of the figure. To achieve more comprehensive knowledge, functional annotations of the molecule are first obtained under different classification systems, as illustrated in Step 3. Then, these annotations are summarized using GPT-4o, resulting in a higher-quality textual description, as depicted in Step 4. The blue and green highlighted sections illustrate the different perspectives between traditional text and HTA text descriptions. For detailed processing steps, please refer to the Appendix A.

## 3.1 Hierarchical Taxonomic Annotation (HTA)

To enable structured, hierarchical molecular representations, we introduce the Hierarchical Taxonomic Annotation (HTA) framework, which organizes molecular functions across multiple classification levels. This setup allows the model to capture fine-grained, hierarchical semantics essential for understanding complex molecular properties and their biological roles. We curate a high-quality dataset of 47,269 *<SMILES, Text, HTA>* triplets sourced from PubChem [13]. As shown in Figure 2, these triplets are annotated across 32 diverse hierarchical classification systems, providing a comprehensive, multi-level understanding of molecular behavior. Figure 2 illustrates the construction pipeline for HTA. Beginning with a molecule's SMILES representation, the molecule is queried against PubChem [13]. This yields a set of traditional functional descriptions, which are typically concise, ontology-aware summaries based on cheminformatics rules. For example, citronellol is described as *a monoterpenoid... with a role as a plant metabolite.* While such descriptors are chemically accurate, they often lack broader context, such as ecological origin, industrial relevance, or toxicological implications.

To address this limitation, we augment the molecule's annotation space through structured taxonomic enrichment by mapping it into multiple biological and chemical taxonomies. For example, the LOTUS Tree [33] highlights natural product classifications, whereas the MeSH (Medical Subject Headings) Tree [14] emphasizes medical functionalities of the same molecule. Through this multi-perspective approach, these hierarchies expand the molecular profile beyond flat descriptors into deeply nested semantic trees spanning chemistry, biology, and pharmacology.

In the final stage, we leverage a GPT-4o [1, 25] to synthesize the retrieved structured annotations into a high-fidelity, human-readable HTA. Unlike traditional descriptors, HTAs encode multi-perspective knowledge: they trace the chemical derivation (e.g., from citronellal), mention natural sources (e.g., rose oil), functional applications (e.g., fragrance in various products), and regulatory or biomedical associations (e.g., environmental protection agencies, blood exposome). This generative synthesis is guided by structural prompts and validated by domain experts to ensure factual accuracy and interoperability.

Crucially, the information content in HTAs is complementary to traditional functional annotations. While the latter provides standardized yet narrow chemical definitions, HTAs integrate cross-domain knowledge that aligns better with how biological and industrial experts interpret molecular function. The results (Table 3) indicate that simultaneously incorporating HTAs and traditional functional

annotations helps the model capture both fine-grained structural features and broader biological semantics, leading to improved performance across a range of molecular property prediction tasks.

## 3.2 Geometry-based Global Alignment

We aim to learn meaningful multimodal representations by jointly modeling three data modalities: molecule SMILES ($M$), textual descriptions ($T$), and HTA ($H$). SMILES representations utilize the encoder $E_m$, while both textual descriptions ($T$) and HTA ($H$) share a common text encoder $E_t$.

Traditional multimodal approaches typically rely on pairwise similarity metrics such as cosine similarity: $\cos(\theta_{ij}) = \frac{\langle M_i, M_j \rangle}{\|M_i\| \cdot \|M_j\|}$. However, these methods often anchor one modality and align others to it independently, failing to capture higher-order relationships across all modalities. To address this, GRAM [5] introduced a geometry-based alignment approach that uses the volume of the parallelotope spanned by modality vectors as a global measure of alignment. Specifically, for three normalized embeddings $m$, $t$, and $h$, the volume of the parallelotope is computed as $\mathrm{Vol}(m, t, h) = \sqrt{1 - \langle m, t \rangle^2 - \langle m, h \rangle^2 - \langle t, h \rangle^2 + 2\langle m, t \rangle \langle t, h \rangle \langle h, m \rangle}$, which reflects the overall geometric alignment of the embeddings. The volume shrinks as the modalities converge and grows as they diverge. Unlike pairwise contrastive learning methods, this formulation was shown to capture the global structure of cross-modal interactions in a principled and scalable way for audio-video-text pairs [5].

**Global Volume-based Contrastive Loss.** Following the approach introduced in GRAM [5], we construct a *global contrastive objective* over the three modalities—SMILES, traditional text descriptions, and HTA annotations. Each modality is processed through a modality-specific encoder followed by a modality-specific projection head (implemented as a three-layer MLP) to map the embeddings into a shared latent space, yielding embeddings $m$, $t$, and $h$ for SMILES, text, and HTA, respectively.

To holistically align the three modalities, we compute the volume of the parallelotope formed by the triplet of unit-normalized vectors $(m, t, h)$. We define a *bidirectional global contrastive loss* that captures two complementary retrieval directions. In the first direction, denoted $\mathcal{L}_{\mathrm{M2TH}}$, the model is trained to retrieve the correct semantic context—comprising both textual and taxonomic annotations—given a molecular embedding. That is, given $m_i$, the loss encourages the volume $\mathrm{Vol}(m_i, t_i, h_i)$ to be smaller than volume spanned by any mismatched triplets $(m_i, t_j, h_j)$ for $j \neq i$:

$$\mathcal{L}_{\mathrm{M2TH}} = -\frac{1}{B} \sum_{i=1}^{B} \log \frac{\exp(-\mathrm{Vol}(m_i, t_i, h_i)/\tau)}{\sum_{j=1}^{B} \exp(-\mathrm{Vol}(m_i, t_j, h_j)/\tau)}, \tag{1}$$

where $B$ is the batch size and $\tau$ is a learnable temperature parameter.

Conversely, the second direction, $\mathcal{L}_{\mathrm{TH2M}}$, considers the retrieval of the correct molecule given the semantic context. Here, the volume of the correct triplet $(m_i, t_i, h_i)$ is minimized relative to all volumes spanned by mismatched triples $(m_j, t_i, h_i)$:

$$\mathcal{L}_{\mathrm{TH2M}} = -\frac{1}{B} \sum_{i=1}^{B} \log \frac{\exp(-\mathrm{Vol}(m_i, t_i, h_i)/\tau)}{\sum_{j=1}^{B} \exp(-\mathrm{Vol}(m_j, t_i, h_i)/\tau)}. \tag{2}$$

The final loss averages both directions to ensure mutual semantic alignment of all three modalities:

$$\mathcal{L}_{\mathrm{g}} = \frac{1}{2}(\mathcal{L}_{\mathrm{M2TH}} + \mathcal{L}_{\mathrm{TH2M}}). \tag{3}$$

This bidirectional formulation encourages robust triadic alignment, capturing global structure across modalities more effectively than traditional pairwise contrastive losses.

## 3.3 Fine-grained Local Alignment

While the global alignment captures the overall semantic relationships among modality embeddings, it may overlook fine-grained correspondences between molecular functional sub-groups and their sub-textual descriptions. For instance, local features such as aromatic rings, hydroxyl groups, or aliphatic chains often correspond to specific phrases in molecular descriptions or to fine-level taxonomic labels.

To address this limitation, we introduce a *local alignment contrastive loss* that complements the global volume-based objective. Unlike GRAM [5], which operates solely at the level of full modality embeddings, our method leverages the compositional nature of molecules to align substructures with their semantic counterparts in text and taxonomy.

By decomposing each molecule into interpretable substructures and anchoring them to matched textual or taxonomic segments, we encourage the model to learn fine-grained correspondences across modalities. This local supervision enforces semantic consistency not only at the global level but also within the internal structure of molecular representations.

### 3.3.1 Functional Group-Level Representation

To enable fine-grained alignment, we construct a high-quality dataset that links functional group structures with their corresponding semantic descriptions. Using RDKit, we screen and identify functionally significant groups frequently found in drug-like molecules. These groups include moieties such as hydroxyls, amines, carboxyls, and aromatic systems. For each group, comprehensive textual descriptions are curated through a hybrid process involving GPT-4o [1] and expert review by professional chemists. This ensures both semantic richness and domain accuracy, resulting in a curated dataset of 85 functional groups paired with high-quality textual annotations.

During training, we extract all $(k)$ prominent functional groups from each molecule based on its SMILES string using the RDKit parser. Each functional group is encoded into the shared latent space using modality-specific encoders: the SMILES encoder and projector are used to obtain the structural embeddings $fg_1, fg_2, \ldots, fg_k$, while the corresponding textual descriptions are processed through the text encoder and projector to produce the text embeddings $fgt_1, fgt_2, \ldots, fgt_k$. To obtain a consolidated representation, we apply a max-pooling operation over the individual embeddings:

$$fg_{\text{pooled}} = \text{Pool}(fg_1, fg_2, \ldots, fg_k), \tag{4}$$

$$fgt_{\text{pooled}} = \text{Pool}(fgt_1, fgt_2, \ldots, fgt_k). \tag{5}$$

This process forms the basis to align sets of fine-grained functional units in molecules with their semantic counterparts in natural language for our local contrastive alignment loss.

### 3.3.2 Local Alignment Loss

Using these pooled representations, we define our bidirectional local alignment contrastive loss as follows.

$$\mathcal{L}_{FG2T} = -\frac{1}{B} \sum_{i=1}^{B} \log \frac{\exp(fg_{pooled,i} \cdot fgt_{pooled,i}/\tau)}{\sum_{j=1}^{B} \exp(fg_{pooled,i} \cdot fgt_{pooled,j}/\tau)},$$

$$\mathcal{L}_{T2FG} = -\frac{1}{B} \sum_{i=1}^{B} \log \frac{\exp(fg_{pooled,i} \cdot fgt_{pooled,i}/\tau)}{\sum_{j=1}^{B} \exp(fg_{pooled,j} \cdot fgt_{pooled,i}/\tau)}, \tag{6}$$

$$\mathcal{L}_{\text{l}} = \frac{1}{2}(\mathcal{L}_{FG2T} + \mathcal{L}_{T2FG}),$$

where $B$ is the batch size and $\tau$ is the same temperature parameter used in the global loss. The local contrastive loss operates bidirectionally to ensure mutual semantic grounding between functional groups and text. The first term, $\mathcal{L}_{\text{FG2T}}$, can be seen as asking: Given a structural embedding of functional groups, can we retrieve the correct description from a pool of candidates? Conversely, the second term, $\mathcal{L}_{\text{T2FG}}$, poses the reverse query: Given a textual description, can we recover the correct functional group structure from a batch of molecules? This dual supervision encourages the model to not only generate chemically meaningful embeddings of functional substructures but also associate them with precise and unambiguous textual counterparts.

### 3.4 Momentum-based Integration

To effectively integrate global and local alignments, we adopt a momentum-based approach that dynamically adjusts the importance of each alignment component:

$$\mathcal{L} = \alpha \mathcal{L}_{\text{g}} + (1 - \alpha)\mathcal{L}_{\text{l}}, \tag{7}$$

where $\alpha$ is a momentum coefficient that balances global and local alignments. Instead of using a fixed $\alpha$, we employ an exponential moving average to update it during training:

$$\alpha_t = \beta \alpha_{t-1} + (1 - \beta) \cdot \frac{\mathcal{L}_{\mathrm{g}}^{(t)}}{\mathcal{L}_{\mathrm{g}}^{(t)} + \mathcal{L}_{\mathrm{l}}^{(t)}}, \tag{8}$$

where $\beta$ is a momentum parameter (0.9), and $\mathcal{L}_{\mathrm{g}}^{(t)}$ and $\mathcal{L}_{\mathrm{l}}^{(t)}$ are the respective loss values at training step $t$. This dynamic adjustment ensures that the model focuses more on the alignment component that currently has higher loss, effectively addressing the most pressing alignment challenges at each training stage.

## 4 Experiments

In this section, we present the main results of the proposed multimodal alignment framework across several downstream molecular property prediction tasks. We aim to assess how well our method leverages information from SMILES strings, textual descriptions, and hierarchical taxonomic annotations. We begin by describing the datasets, tasks, and baselines used in our study, followed by a discussion of the main results. Finally, we provide an ablation study to isolate the contribution of each component in our framework. A detailed experimental setup can be found in the Appendix B.

**Pre-training Datasets.** The pre-training dataset is sourced from PubChem, initially following the method described in [16] to obtain approximately 380k SMILES-text pairs. After a series of filtering steps (for detailed processing steps, please refer to the Appendix A) and obtaining HTA information for each molecule, our final pre-training dataset consists of 47,269 SMILES-Text-HTA triplets. The annotations cover various biological roles, molecular functions, mechanisms of action, and multi-level bioactivity information.

**Molecular property prediction benchmarks.** We evaluate our model on a broad range of molecular property prediction tasks drawn from two major benchmarks: MoleculeNet [37] and the Therapeutics Data Commons (TDC) [10]. For MoleculeNet, we include 8 classification datasets and 3 regression datasets. The classification tasks comprise toxicity prediction (BBBP, Tox21, ToxCast), side-effect and clinical toxicity prediction (Sider, ClinTox), and bioactivity classification (MUV, HIV, Bace), with performance reported using the ROC-AUC metric. The regression tasks include molecular solubility (ESOL), solvation free energy (FreeSolv), and lipophilicity (Lipophilicity), with performance reported using RMSE.

For the TDC benchmark, we evaluate on 7 datasets, including 6 classification datasets (DILI, Carcinogens (Languin), Skin Reaction, hERG, AMES, and CYP P450 2C19) and 1 regression dataset (Caco-2). For classification tasks, we report both AUC and accuracy following TDC guidelines, while for the regression task, we report RMSE. Following standard practice, we adopt the *scaffold split* throughout our methodology to evaluate generalization to novel chemical scaffolds. Each experiment is repeated across three random seeds, and we report the mean and standard deviation. A detailed dataset description can be found in Appendix C.

Table 1: Performance comparison on molecule property prediction. We present the ROC-AUC(%) scores of the molecular property prediction task on MoleculeNet. For baselines that report results, we directly use their reported outcomes. Note that MolCA-SMILES does not report results for the MUV and HIV datasets. The best results are marked in **bold**, and the second-best are underlined.

| Method | BBBP | Tox21 | ToxCast | Sider | ClinTox | MUV | HIV | Bace | Avg |
|---|---|---|---|---|---|---|---|---|---|
| MOLFORMER | 70.74±1.34 | 74.74±0.56 | 65.51±0.63 | 61.75±1.23 | 77.64±0.98 | 67.58±1.01 | 75.64±1.76 | 78.64±2.35 | 71.53 |
| KV-PLM | 70.50±0.54 | 72.12±1.02 | 55.03±1.65 | 59.83±0.56 | 89.17±2.73 | 54.63±4.81 | 65.40±1.69 | 75.80±2.73 | 67.81 |
| MegaMolBART | 68.89±0.17 | 73.89±0.67 | 63.32±0.79 | 59.52±1.79 | 78.12±4.62 | 61.51±2.75 | 71.04±1.70 | 82.46±0.84 | 69.84 |
| MoleculeSTM-SMILES | 70.75±1.90 | 75.71±0.89 | 65.17±0.37 | 63.70±0.81 | 86.60±2.28 | 65.69±1.46 | 77.02±0.44 | 81.99±0.41 | 73.33 |
| MolFM | 72.90±0.10 | 77.20±0.70 | 64.40±0.20 | 64.20±0.90 | 79.70±1.60 | 76.00±0.80 | 78.80±1.10 | 83.90±1.10 | 74.64 |
| MoMu | 70.50±2.00 | 75.60±0.30 | 63.40±0.50 | 60.50±0.90 | 79.90±4.10 | 70.50±1.40 | 75.90±0.80 | 76.70±2.10 | 71.63 |
| Atomas | 73.72±1.67 | 77.88±0.36 | 66.94±0.90 | 64.40±1.90 | 93.16±0.50 | 76.30±0.70 | 80.55±0.43 | 83.14±1.71 | 77.01 |
| MolCA-SMILES | 70.80±0.60 | 76.00±0.50 | 56.20±0.70 | 61.10±1.20 | 89.00±1.70 | - | - | 79.30±0.80 | 72.10 |
| **TRIDENT (M-S)** | 73.14±0.44 | 78.23±0.12 | 67.79±0.56 | **64.62±0.47** | **95.75±0.71** | 82.88±1.41 | 79.64±1.15 | **84.19±0.95** | 78.28 |
| **TRIDENT (M-M)** | **73.95±1.01** | **79.36±0.13** | **67.80±0.37** | 63.64±0.56 | 95.41±0.66 | **83.51±0.48** | **81.63±0.52** | 82.39±0.56 | **78.46** |

Table 2: Performance of different methods on DILI, Carcinogens, and Skin Reaction tasks, reporting AUC and Accuracy. The best results are marked in **bold**, and the second-best are underlined.

| Method | DILI (475 drugs) | | Carcinogens (278 drugs) | | Skin Reaction (404 drugs) | |
|---|---|---|---|---|---|---|
| | AUC | ACC | AUC | ACC | AUC | ACC |
| MOLFORMER | 85.59±1.39 | 76.39±5.24 | 77.27±0.76 | 77.32±1.47 | 63.75±1.41 | 60.98±3.44 |
| KV-PLM | 73.46±0.61 | 62.50±2.08 | 75.18±3.71 | 76.01±1.75 | 62.88±2.30 | 59.76±5.17 |
| MolT5 | 77.37±1.15 | 69.44±1.20 | 86.89±1.00 | 84.45±1.11 | 68.67±3.99 | 62.22±1.41 |
| MoMu | 80.44±2.47 | 75.00±4.17 | 80.11±1.50 | 78.00±2.62 | 61.63±1.94 | 56.10±3.45 |
| MolCA-SMILES | 88.34±1.28 | 80.56±2.40 | 82.00±1.80 | 78.76±0.52 | 65.13±0.88 | 62.20±1.72 |
| MoleculeSTM-SMILES | 91.20±2.02 | 84.72±2.41 | 83.87±1.30 | 81.05±0.63 | 67.72±0.50 | 61.60±0.73 |
| MolXPT | 91.67±0.76 | 84.03±3.19 | 75.76±2.73 | 80.90±2.06 | 61.08±1.28 | 62.60±1.40 |
| BioT5 | 82.45±1.81 | 76.39±3.18 | 82.83±4.31 | 76.19±2.06 | 68.27±4.41 | 62.21±1.06 |
| BioT5+ | 82.58±1.65 | 80.56±1.20 | 86.62±2.32 | 77.41±2.00 | 65.25±0.66 | 62.27±1.20 |
| Atomas | 90.17±1.30 | 85.08±2.16 | 82.47±2.11 | 80.75±0.50 | 70.33±0.88 | 61.79±6.14 |
| **TRIDENT (M-S)** | **95.08±0.70** | **86.81±2.40** | 83.42±1.10 | 81.47±0.92 | 70.33±0.63 | **63.42±4.22** |
| **TRIDENT (M-M)** | 94.56±0.88 | 86.80±3.18 | **87.07±0.77** | **84.62±1.07** | **72.00±1.09** | 62.60±1.40 |

**Baselines.**   We compare our TRIDENT approach against a range of recent state-of-the-art baselines. These include transformer-based models that use SMILES representations, such as MOL-FORMER [31], MegaMolBART [29], MolXPT [18], BioT5 [27], and BioT5+[26]. We also compare against multimodal approaches incorporating additional textual and molecular information, including MolFM[21], MoMu [35], MoleculeSTM [16], MolCA-SMILES [19], KV-PLM [39], and Atomas [41]. Additionally, we compare with Uni-Mol [42], which employs 3D molecular conformations for representation learning. A detailed baseline introduction can be found in Appendix D.

## 4.1 Results and Analysis

**Molecular property prediction.**   As shown in Table 1, our TRIDENT model achieves state-of-the-art performance across the diverse set of MoleculeNet tasks, consistently outperforming prior methods. We evaluate two encoder configurations: TRIDENT (M-S), which uses MOLFORMER [31] as the SMILES encoder and SciBERT [2] as the text encoder, and TRIDENT (M-M), which combines MOLFORMER with MolT5 [7] as text encoder. On average, TRIDENT (M-M) achieves the highest ROC-AUC score of 78.5%, outperforming strong baselines such as Atomas (77.01%) and MolFM (74.62%). It achieves best-in-class performance on 5 of the 8 tasks, including challenging benchmarks such as BBBP, Tox-21, Toxcast, MUV, and HIV. The M-S variant is also highly competitive, outperforming nearly all baselines and obtaining the top score on Bace, Sider, and ClinTox.

Furthermore, we evaluate our method on MoleculeNet regression tasks, as shown in Appendix E. TRIDENT (M-M) consistently demonstrates superior performance, achieving the best or competitive RMSE scores across all three regression datasets. These results further validate the effectiveness of our multimodal learning approach.

One reason TRIDENT performs best is that most prior methods rely solely on generic textual descriptions of molecular function, lacking the multi-dimensional, hierarchical annotations provided by our HTA dataset. Furthermore, the vast majority of approaches overlook the importance of local alignment between molecular subgraphs and their corresponding textual fragments. While Atomas does introduce a local alignment component via attention, that static attention scheme cannot dynamically balance the competing demands of global context and fine-grained substructure matching throughout training. In our framework, we integrate a momentum-based alignment mechanism that

Table 3: Performance comparison of molecular property prediction methods based on different input modalities (SMILES, Text, and HTA) across various datasets (ROC-AUC%). Best results in **bold**.

| Method | Input | | | Datasets | | | | |
|---|---|---|---|---|---|---|---|---|
| | SMILES | Text | HTA | BBBP | Tox21 | ToxCast | Sider | Bace |
| TRIDENT (M-M) | ✓ | ✗ | ✓ | 72.02±0.36 | 78.21±0.19 | 67.04±0.38 | 63.18±0.31 | 81.28±0.92 |
| TRIDENT (M-M) | ✓ | ✓ | ✓ | **73.95±1.01** | **79.36±0.13** | **67.80±0.37** | **63.64±0.56** | **82.39±0.56** |

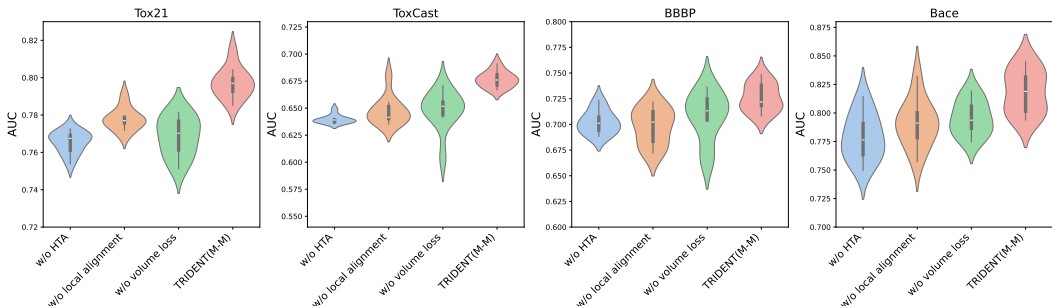

Figure 3: The ablation experiments are conducted on the Tox21, ToxCast, BBBP and Bace datasets. "`w/o HTA`" denotes that only not use hierarchical taxonomic annotation; "`w/o local alignment`" denotes that the local alignment is removed; and "`w/o volume loss`" indicates that only the volume-based loss is changed to the standard contrastive loss.

adaptively reweights global and local objectives. Our experiments show that combining HTA with this dynamic balancing of global and local alignment yields substantial improvements across a wide array of molecular property prediction tasks.

To further validate the effectiveness of our model, we select three small datasets from TDC. This choice is motivated by the challenging nature of data acquisition for toxicity, making small datasets more reflective of the model's ability to generalize and perform well in scenarios with limited data. By evaluating on these data-scarce tasks, we aim to demonstrate the robustness and adaptability of our approach in real-world settings. As shown in Table 9, TRIDENT again achieves new state-of-the-art results across all three datasets. TRIDENT (M-S) obtains the highest AUC and accuracy scores on DILI, alongside strong performance on Carcinogens and Skin Reaction. The TRIDENT (M-M) variant further improves AUC on Carcinogens and Skin Reaction, outperforming all baselines. Notably, TRIDENT excels not only on these smaller datasets but also demonstrates superior performance on larger-scale benchmarks, including AMES (7,255 drugs), CYP P450 2C19 (12,665 drugs), and the regression dataset Caco-2 (906 drugs) (see Appendix E). These results highlight the broad applicability of TRIDENT across diverse molecular property prediction tasks, ranging from data-scarce to data-rich settings.

## 4.2 Ablation Study

To understand the contribution of different components in our TRIDENT framework, we conduct a detailed ablation study. We compare several model variants on representative tasks to disentangle the impact of local functional group and sub-textual description alignment loss, hierarchical taxonomic supervision as well as the volume loss for global alignment.

As shown in Figure 3, removing HTA information (w/o HTA) leads to a noticeable drop in model performance, highlighting the importance of HTA in capturing a rich, multi-level understanding of molecular behavior through hierarchical taxonomy annotations. Similarly, excluding the local-alignment component (w/o local alignment) results in a clear performance decline, showing how fine-grained alignment plays a critical role in enhancing the model's capability. Interestingly, replacing our volume loss with standard contrastive loss (w/o volume loss) causes significant instability on datasets

Table 4: Strategies for combining global and local loss functions (ROC-AUC%). Sum: direct addition; Curve: sigmoid-weighted combination with increasing local loss weight; Momentum: dynamic alignment approach. Best results in **bold**.

| Method | Tox21 | ToxCast | BBBP | Bace |
|---|---|---|---|---|
| Sum | 77.79±0.81 | 66.73±0.65 | 72.15±0.81 | 81.42±0.69 |
| Curve | 76.68±0.79 | 65.49±0.82 | 71.68±0.79 | 80.91±0.83 |
| **Momentum** | **79.36±0.13** | **67.80±0.37** | **73.95±1.01** | **82.39±0.56** |

like Tox21, ToxCast, and BBBP. This is likely because traditional alignment approaches struggle to handle multiple modalities effectively [5]. In addition, our momentum-based mechanism further strengthens generalization by dynamically balancing global and local objectives during training, as demonstrated in Table 4. Overall, the full TRIDENT framework consistently outperforms all ablated versions, confirming the value and necessity of each individual component.

In addition, to further explore the relationship between HTA and general molecular descriptions, we directly use HTA and molecular SMILES as inputs for the global module during pretraining, replacing the volume loss with standard contrastive learning loss while keeping other settings unchanged. The results are shown in Table 3. When using HTA as the sole text input, the model already outperforms most baselines but still falls short of the tri-modal input. This may be because HTA text and traditional molecular descriptions complement each other in terms of information representation. HTA text contains up to 32 categorical annotations, providing more diverse and multi-angled molecular information, while traditional functional descriptions are more direct and highlight the core features of molecular structures. Therefore, by simultaneously leveraging HTA text and traditional descriptions as multimodal inputs, the model captures molecular characteristics more comprehensively, thereby further improving its performance.

## 5  Conclusion

TRIDENT is a tri-modal molecular representation framework that unifies chemical SMILES, natural-language descriptions, and hierarchical taxonomic annotations into a single, semantically rich embedding space. Trained on over 47,269 *<SMILES, Text, HTA>* triplets, it uses a geometry-aware volume-based contrastive loss for global alignment and a local contrastive module for precise substructure–text matching, overcoming modality misalignment and flat representations. A momentum-based weighting scheme balances global and local objectives, delivering state-of-the-art performance on 11 property-prediction benchmarks without altering the architecture. These results highlight the power of structurally and semantically grounded multimodal alignment in molecular learning. More broadly, this work underscores the importance of hierarchical, multi-resolution reasoning in molecular modeling and opens new directions for scalable, and biologically meaningful representation learning in the chemical sciences. One limitation of our work is that molecular properties such as toxicity depend not only on molecular structure but also on targets and metabolites, which are not currently captured and slated for future research.

## 6  Acknowledgments

This work was partially supported by a Johnson & Johnson grant for LLM-based toxicity prediction, US National Science Foundation IIS-2412195, CCF-2400785, the Cancer Prevention and Research Institute of Texas (CPRIT) award (RP230363), the National Institutes of Health (NIH) R01 award (1R01AI190103-01) and Microsoft Accelerate Foundation Models Research (2024).

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

# Appendix

## A    Data Collection and Processing

We obtain a dataset containing 320,000 molecule-text pairs from the PubChem database and preprocess the text descriptions following the MolecularSTM method. Specifically, molecule names are replaced with "this molecule is..." or "these molecules are..." to prevent the model from recognizing molecules based solely on their names. Additionally, to create unique SMILES-text pairs, we merge molecules with the same CID (chemical identifier) and filter out text descriptions with fewer than 18 characters.

Moreover, we use PubChem's classification system to obtain up to 32 classification descriptions for each molecule, as illustrated in Algorithm 1. Ultimately, we generate 47,269 <SMILES, Text, Hierarchical Taxonomic Annotation> triplets. As shown in Figure 4, to further optimize and summarize the classification annotations, we use GPT-4 to generate summarized descriptions, resulting in high-quality HTA text descriptions.

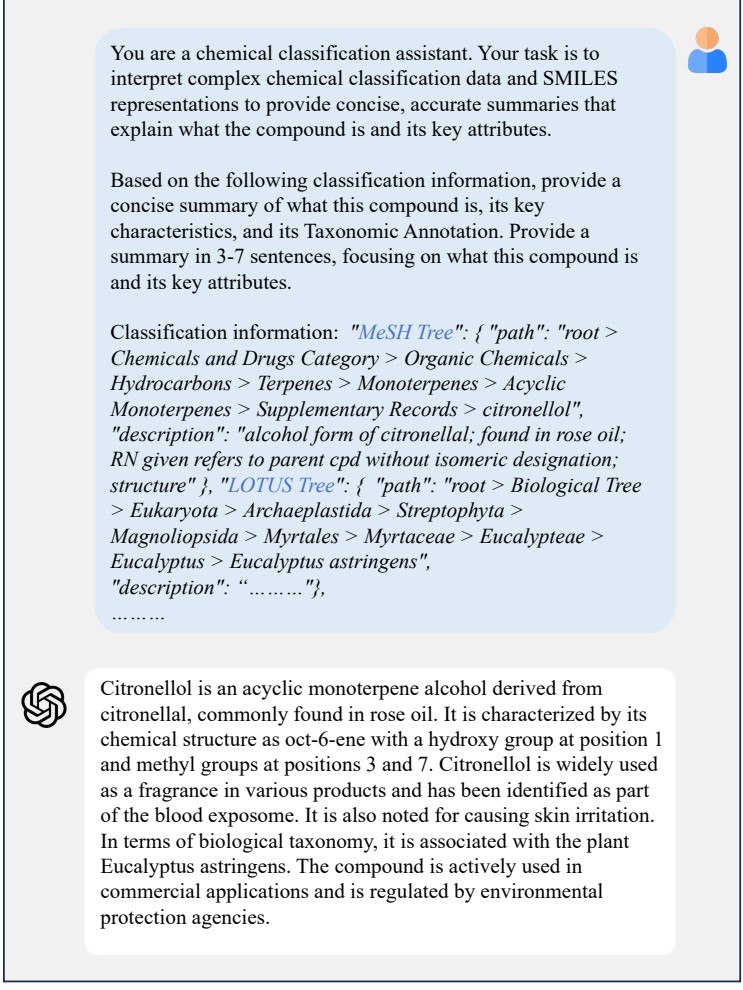

Figure 4: The workflow for summarizing Hierarchical Taxonomic Annotations (HTA). Using GPT-4o, detailed classification annotations are processed and summarized, resulting in high-quality HTA text descriptions for molecular data.

**Algorithm 1:** Hierarchical Taxonomic Annotation: retrieval of molecule classification from PubChem.

---

1 **Input preparation**
2     Load CID list . . . . . . . . . . . . . . . . . . . . . . . . . {from CIDs.txt file}
3     Configure batch size . . . . . . . . . . . . . . . . . . . . . . {batch_size = 20}

4 **Batch processing setup**
5     Thread pool executor . . . . . . . . . . . . . . . . . . . . . {max_workers = 5}
6     Retry mechanism . . . . . . . . . . . . . . {max_retries = 3, backoff_factor = 2}
7     Rate limiting . . . . . . . . . . . . . . . . {0.5s between API calls, 3s batches}

8 **API interaction**
9     Classification headers retrieval
10        Endpoint {PubChem /pug_view/data/compound/{cid}/JSON/?heading=Classification}
11        Output . . . . . . . . . . . . . . . . . {list of classification systems with HIDs}
12     Classification path retrieval
13        Endpoint . . . . . . {PubChem /classification_2.fcgi?hid={hid}&search_uid={cid}}
14        Path construction . . . . . . . . . . . . {recursive traversal of parent-child nodes}
15        Output . . . . . . . . . . . . . . . . . . {hierarchical path string, description}

16 **Result processing**
17     Data structure . . . . . . . . . {CID $\rightarrow$ {classification_system $\rightarrow$ {path, description}}}
18     Intermediate saving . . . . . . . . . . . . . . {save after each batch for resumability}
19     Error handling . . . . . . . . . . . . . {log warnings and errors, continue processing}

20 **Output**
21     JSON file . . . . . . . . . . . . . . . {complete taxonomy annotation for all CIDs}

---

## B   Experimental Setup

### B.1   Model Architecture

As shown in Algorithm 2, our multimodal contrastive learning framework consists of the following key components:

1. **Three-modal encoding**: MoLFormer processes SMILES structures, while SciBERT encodes molecular text descriptions and category information, outputting 768-dimensional features.

2. **Feature projection**: Multi-layer MLPs project features from each modality into a 512-dimensional shared space with L2 normalization.

3. **Two-level contrastive learning**:
   - Global contrast: Applies GRAM3Modal method to calculate volume loss and InfoNCE loss across three modalities.
   - Local contrast: Aligns SMILES and text representations at the functional group level

4. **Dynamic loss integration**: Employs a momentum update mechanism ($\beta = 0.9$) to adaptively adjust weights between global and local losses, with total loss $L = \alpha \cdot L_{\text{global}} + (1 - \alpha) \cdot L_{\text{local}}$.

The model is implemented in a distributed training environment, freezing pre-trained encoders and optimizing only projection layer parameters.

### B.2   Training Configuration

Our multimodal contrastive learning model was trained on two NVIDIA H100 GPUs with the following configuration, as shown in Table 5.

Each training epoch takes approximately 5 minutes. During training, we used DistributedSampler to ensure consistent data distribution across different GPUs and shuffled the data by setting different random seeds at the beginning of each epoch. Due to the large size of MoLFormer and SciBERT

**Algorithm 2:** MultiModal Contrastive Learning: three-modal alignment with momentum integration.

---

1  **Input modality encoders**
2      SMILES encoder ...................... {MoLFormer (768-dim)}
3      Text description encoder ................... {SciBERT (768-dim)}
4      Category encoder ................ {shared with text encoder (768-dim)}

5  **Projection layers**
6      SMILES projection
          {Linear→GELU→LayerNorm→Dropout→Linear→GELU→LayerNorm→Linear
          (512-dim)}
7      Text projection
          {Linear→GELU→LayerNorm→Dropout→Linear→GELU→LayerNorm→Linear
          (512-dim)}

8  **Global contrastive loss (GRAM3Modal)**
9      Volume computation ................. {determinant of Gram matrix}
10     Temperature scaling ......................... $\{\tau = 0.07\}$
11     Volume-based alignment ............. {cross entropy on negative volumes}
12     InfoNCE alignment ............. {standard contrastive across modalities}

13 **Local functional group alignment**
14     Functional group detection ................... {RDKit Fragments}
15     FG representation ............ {weighted pooling of fragment embeddings}
16     FG contrastive loss ............ {local InfoNCE between SMILES and text}

17 **Momentum-based loss integration**
18     Momentum coefficient ......................... $\{\beta = 0.9\}$
19     Initial alpha ............................... $\{\alpha = 0.5\}$
20     Dynamic update ........ $\{\alpha = \beta \cdot \alpha_{prev} + (1 - \beta) \cdot (global\_loss/total\_loss)\}$

21 **Training configuration**
22     Optimization .......................... {Adam (lr=1e-5)}
23     Encoder freezing ................. {both text and SMILES encoders}
24     Distributed training ..................... {DDP, NCCL backend}
25     Batch size ........................ {40 per GPU, multi-GPU}

---

models, we adopted a strategy of freezing pre-trained encoder parameters and only training projection layer parameters, which significantly reduced computation and memory requirements while maintaining model expressiveness. We observe that the dynamic integration of global and local losses (dynamic adjustment of $\alpha$ value) demonstrates good adaptability during the training process, enabling reasonable balancing of the contributions from the two losses at different training stages.

## B.3   Evaluation Metrics

To comprehensively evaluate the performance of our multimodal contrastive learning model on molecular property prediction tasks, we adopt appropriate evaluation metrics based on the characteristics of different datasets.

### B.3.1   MoleculeNet Datasets

For binary classification tasks in MoleculeNet datasets, we employ ROC-AUC (Receiver Operating Characteristic Area Under Curve) and standard deviation as the primary evaluation metric. The ROC-AUC is calculated as follows:

$$\text{AUC} = \int_0^1 \text{TPR}(\text{FPR}^{-1}(t)) \, dt \tag{9}$$

Table 5: Training Configuration Details

| Parameter | Configuration |
|---|---|
| Hardware environment | $2 \times$ NVIDIA H100 GPU |
| Training framework | PyTorch DistributedDataParallel (DDP) |
| Communication backend | NCCL |
| Optimizer | Adam |
| Learning rate | 1e-5 |
| Batch size | 40 per GPU (total batch size = 80) |
| Weight decay | 1e-4 |
| Training epochs | 60 epochs |
| Training dataset size | 47,269 molecule-text-HTA pairs |
| Gradient accumulation steps | 1 |
| Learning rate schedule | Fixed learning rate, no decay |
| Early stopping | Stop after 5 epochs without validation loss improvement |
| **Loss Function Configuration** | |
| Contrastive temperature | $\tau = 0.07$ |
| Momentum coefficient | $\beta = 0.9$ |
| Initial loss weight | $\alpha = 0.5$ |
| Global loss composition | GRAM3Modal volume loss + InfoNCE loss |
| Local loss composition | Functional group level InfoNCE loss |
| Label smoothing parameter | 0.1 |
| **Model Configuration** | |
| Modality encoders | Frozen (feature extraction only) |
| Projection layers | Fully fine-tuned (768-dim $\rightarrow$ 512-dim) |
| Dropout rate | 0.1 |
| Gradient clipping | Max norm 1.0 |
| Mixed precision training | FP16 |
| Checkpoint saving frequency | Every 2 epochs |

where the True Positive Rate (TPR) and False Positive Rate (FPR) are defined as:

$$\text{TPR} = \frac{\text{TP}}{\text{TP} + \text{FN}} \tag{10}$$

$$\text{FPR} = \frac{\text{FP}}{\text{FP} + \text{TN}} \tag{11}$$

ROC-AUC values range from 0 to 1, with values closer to 1 indicating better model performance. This metric demonstrates good robustness to class imbalance issues, making it particularly suitable for molecular property prediction tasks in the pharmaceutical domain where positive and negative samples are often unevenly distributed.

$$\text{STD} = \sqrt{\frac{1}{n-1} \sum_{i=1}^{n} (x_i - \bar{x})^2} \tag{12}$$

where $n$ is the number of experiments, $x_i$ is the result of the $i$-th experiment, and $\bar{x}$ is the mean of $n$ experiments. The standard deviation reflects the stability and reliability of model performance, with smaller standard deviations indicating more stable performance across different data splits and random seeds.

Table 6: MoleculeNet Datasets Details

| Dataset | Sample Size | Prediction Task | Task Description |
|---|---|---|---|
| BBBP | 2,050 | Blood-Brain Barrier Penetration | Predicts whether compounds can penetrate the blood-brain barrier |
| Tox21 | 7,831 | Toxicity Assessment | Evaluates compound activity across 12 different toxicity pathways |
| ToxCast | 8,597 | Toxicity Prediction | Predicts compound toxicity across 617 biological assays |
| SIDER | 1,427 | Side Effect Prediction | Predicts adverse drug reactions covering 27 types of side effects |
| ClinTox | 1,483 | Clinical Toxicity | Evaluates clinical toxicity and FDA approval status of compounds |
| MUV | 93,087 | Biological Activity | Molecular activity prediction with 17 highly imbalanced biological targets |
| HIV | 41,127 | Antiviral Activity | Predicts compound inhibition of HIV replication |
| BACE | 1,513 | Enzyme Inhibition | Predicts $\beta$-secretase inhibitor activity for Alzheimer's disease drug discovery |
| ESOL | 1,127 | Water Solubility | Predicts aqueous solubility (log S), fundamental for drug formulation and delivery |
| FreeSolv | 641 | Solvation Free Energy | Predicts hydration free energy, important for understanding molecular interactions |
| Lipophilicity | 4,200 | Lipophilicity | Predicts octanol-water partition coefficient (log D), crucial for drug absorption and distribution |

Table 7: TDC Datasets Details

| Dataset | Sample Size | Prediction Task | Task Description |
|---|---|---|---|
| DILI | 475 | Liver Injury Prediction | Predicts drug-induced liver injury, a critical safety consideration in drug development |
| Carcinogens | 278 | Carcinogenicity Prediction | Predicts compound carcinogenicity, crucial for drug and chemical safety evaluation |
| Skin Reaction | 404 | Skin Reaction Prediction | Predicts whether compounds cause skin reactions, important for topical drug development |
| AMES | 7,255 | Mutagenicity Prediction | Predicts compound mutagenicity based on Ames test, standard method for genetic toxicity |
| hERG | 648 | Cardiotoxicity Prediction | Predicts compound blocking activity against hERG potassium channels, major cause of cardiotoxicity |
| CYP P450 2C19 | 12,665 | Drug Metabolism Prediction | Predicts inhibition of CYP2C19 enzyme, essential for assessing drug-drug interactions |
| Caco-2 | 906 | Permeability Prediction | Predicts intestinal permeability using Caco-2 cell line, critical for oral drug bioavailability |

### B.3.2 TDC Datasets

For TDC (Therapeutics Data Commons) datasets, we employ both ROC-AUC and Accuracy as evaluation metrics:

1. **ROC-AUC**: Same definition as in MoleculeNet datasets, used to measure the model's classification performance and discriminative ability.

2. **Accuracy**: The accuracy is calculated as:

$$\text{Accuracy} = \frac{\text{TP} + \text{TN}}{\text{TP} + \text{TN} + \text{FP} + \text{FN}} \tag{13}$$

where TP, TN, FP, and FN represent the number of true positives, true negatives, false positives, and false negatives, respectively.

Accuracy intuitively reflects the proportion of correctly predicted samples by the model. When used in combination with ROC-AUC, it provides a more comprehensive evaluation of model performance. ROC-AUC primarily focuses on the model's ranking ability and threshold-independent performance, while accuracy directly reflects the model's classification effectiveness under specific thresholds.

## C Downstream Tasks Datasets

To comprehensively evaluate the performance of our proposed multimodal contrastive learning framework on molecular property prediction tasks, we conduct extensive experiments on two major benchmark dataset collections: MoleculeNet and TDC (Therapeutics Data Commons).

### C.1 MoleculeNet Datasets

MoleculeNet is one of the most authoritative benchmark dataset collections in the field of molecular machine learning, specifically designed to evaluate the performance of molecular property prediction methods. Table 6 summarizes the detailed information of the 8 MoleculeNet datasets we used.

### C.2 TDC Datasets

TDC (Therapeutics Data Commons) is a large-scale dataset collection specifically designed for therapeutics research, providing more challenging and practically valuable molecular property prediction tasks. Table 7 presents the detailed information of the 7 TDC datasets we selected.

These datasets cover key property prediction tasks in the drug discovery process, including pharmacokinetics (ADME), toxicity, and biological activity across multiple aspects. Both dataset collections are characterized by diversity, challenging nature, standardization, and authority, and are widely recognized and used by both academia and industry.

# D Baselines

In this section, we provide descriptions of the baseline methods used for comparison in our experiments. These baselines represent current approaches in molecular representation learning and multimodal molecular modeling.

## D.1 Single-Modal Baselines

**MOLFORMER**: A transformer-based model that processes SMILES string representations using masked language modeling. The model employs linear attention with rotary positional embeddings and is pre-trained on 1.1 billion molecules from PubChem and ZINC databases in an unsupervised fashion. **MegaMolBART**: A BART-based encoder-decoder model adapted for molecular data. It processes SMILES representations and applies bidirectional and auto-regressive transformers for molecular understanding and generation tasks.

## D.2 Multimodal Baselines

**MoleculeSTM**: A bi-modal model with separate encoders for molecular structures (SMILES/graphs) and textual descriptions. It uses contrastive learning to align structure-text pairs and is trained on over 280,000 molecule-text pairs from PubChem. **MoMu**: A multimodal foundation model that uses separate encoders for molecular graphs and natural language text. The model employs contrastive learning to bridge molecular structures with textual descriptions using paired molecule-text datasets. **MolFM**: A tri-modal model that integrates molecular structures (2D graphs), biomedical texts, and knowledge graphs. It uses cross-modal attention mechanisms and is pre-trained with four objectives: structure-text contrastive learning, cross-modal matching, masked language modeling, and knowledge graph embedding. **KV-PLM**: A BERT-based unified framework that processes both SMILES-encoded molecular structures and natural language text through masked language modeling pre-training. The system enables cross-modal understanding between molecular structures and biomedical text. **MolCA-SMILES**: A molecular graph-language model that uses a Q-Former as a cross-modal projector to bridge graph encoders and language models. The approach employs LoRA adapters and follows a three-stage training pipeline for efficient fine-tuning. **Atomas**: A hierarchical alignment framework that introduces Adaptive Polymerization Module (APM) and Weighted Alignment Module (WAM) to learn fine-grained correspondences between SMILES and text at atom, fragment, and molecule levels. It uses a unified encoder and end-to-end training for joint alignment and generation.

## D.3 Comparison with TRIDENT

Our proposed TRIDENT framework differs from these baselines in several key aspects:

1. **Hierarchical Taxonomic Annotations**: Unlike existing methods that rely on generic textual descriptions, TRIDENT incorporates structured, multi-level functional annotations across 32 taxonomic classification systems, providing richer semantic understanding.

2. **Tri-modal Architecture**: While most baselines focus on bi-modal alignment (structure-text), TRIDENT introduces a novel tri-modal approach that jointly models SMILES, textual descriptions, and hierarchical taxonomic annotations.

3. **Volume-based Global Alignment**: Instead of traditional pairwise contrastive learning, TRIDENT employs a geometry-aware volume-based alignment objective that captures higher-order relationships across all three modalities simultaneously.

4. **Local-Global Integration**: TRIDENT uniquely combines global tri-modal alignment with fine-grained local alignment between molecular substructures and their corresponding textual descriptions, balanced through a momentum-based mechanism.

5. **Dynamic Alignment Strategy**: The momentum-based integration of global and local objectives allows TRIDENT to adaptively focus on different alignment components during training, leading to more robust representation learning.

These innovations enable TRIDENT to achieve state-of-the-art performance across 11 downstream molecular property prediction tasks, demonstrating the effectiveness of our comprehensive multimodal approach.

Table 8: Regression performance (RMSE) on MoleculeNet benchmark. Lower values indicate better performance.The best results are marked in **bold**, and the second-best are underlined.

| Dataset | Uni-Mol | BioT5 | BioT5+ | MolXPT | MolFormer | MolT5 | TRIDENT(M-M) |
|---|---|---|---|---|---|---|---|
| ESOL | $0.79 \pm 0.03$ | $0.80 \pm 0.02$ | $0.79 \pm 0.01$ | $\underline{0.75 \pm 0.01}$ | $0.78 \pm 0.12$ | $0.82 \pm 0.02$ | $\mathbf{0.72 \pm 0.07}$ |
| FreeSolv | $\underline{1.48 \pm 0.05}$ | $1.63 \pm 0.02$ | $1.98 \pm 0.13$ | $1.60 \pm 0.03$ | $1.67 \pm 0.06$ | $1.55 \pm 0.14$ | $\mathbf{1.42 \pm 0.03}$ |
| Lipophilicity | $\mathbf{0.60 \pm 0.02}$ | $0.74 \pm 0.07$ | $0.74 \pm 0.06$ | $0.69 \pm 0.01$ | $0.63 \pm 0.02$ | $0.65 \pm 0.04$ | $\underline{0.60 \pm 0.01}$ |

Table 9: Performance of different methods on AMES and hERG tasks, reporting AUC and Accuracy. The best results are marked in **bold**, and the second-best are underlined.

| Method | AMES (7,255 drugs) | | hERG (648 drugs) | |
|---|---|---|---|---|
| | AUC | ACC | AUC | ACC |
| MOLFORMER | 83.20±0.32 | 78.05±0.76 | 79.65±1.19 | $\underline{81.82\pm3.03}$ |
| KV-PLM | 78.23±0.90 | 71.70±0.94 | 75.87±2.76 | 75.30±3.08 |
| MolT5 | 76.93±0.84 | 70.87±2.22 | 76.25±1.22 | 77.04±4.90 |
| MoMu | 77.20±0.85 | 70.78±0.36 | 75.68±1.89 | 73.27±3.55 |
| MolCA-SMILES | 77.62±1.49 | 71.74±1.07 | 78.40±1.84 | 73.94±4.38 |
| MoleculeSTM-SMILES | 83.60±1.00 | 77.68±0.64 | 79.46±4.63 | 79.19±4.94 |
| MolXPT | 76.93±0.84 | 70.87±2.22 | 82.44±2.14 | 81.31±2.31 |
| BioT5 | 77.57±0.69 | 73.25±1.67 | 77.48±1.59 | 80.30±3.03 |
| BioT5+ | 78.18±1.48 | 73.30±1.15 | 82.27±3.29 | 80.31±1.52 |
| Atomas | 82.63±0.72 | 77.32±0.83 | $\underline{83.34\pm1.79}$ | 78.02±2.00 |
| **TRIDENT (M-S)** | $\underline{85.37\pm0.30}$ | $\underline{78.74\pm0.50}$ | **87.60±1.20** | 81.11±2.64 |
| **TRIDENT (M-M)** | **86.87±0.60** | **80.20±1.44** | 83.31±1.63 | **83.33±2.62** |

# E  Additional Results

In this section, we present additional experimental results that complement the main findings reported in the paper. These include performance evaluations on MoleculeNet regression tasks, larger-scale datasets from the TDC benchmark, more extensive ablation experiments, and additional analyses that provide deeper insights into TRIDENT's capabilities.

## E.1  Performance on MoleculeNet Regression Tasks

To further demonstrate TRIDENT's effectiveness across different task types, we evaluate our method on three regression benchmarks from MoleculeNet: ESOL (water solubility), FreeSolv (solvation free energy), and Lipophilicity (octanol-water partition coefficient). As shown in Table 8, TRIDENT (M-M) achieves the best performance on ESOL and FreeSolv, and matches the state-of-the-art performance on Lipophilicity. These results demonstrate that TRIDENT's multimodal learning framework is effective not only for classification tasks but also for continuous property prediction.

## E.2  Performance on Larger TDC Datasets

While the main paper focused on smaller TDC datasets to demonstrate TRIDENT's data efficiency, we also evaluated our method on larger-scale molecular property prediction tasks. Table 9 presents the results on the AMES mutagenicity dataset (7,255 molecules) and the hERG cardiotoxicity dataset (648 molecules). Additionally, Table 10 reports performance on the CYP P450 2C19 inhibition dataset (12,665 molecules) and the Caco-2 permeability regression dataset (906 molecules).

In summary, TRIDENT's superior performance across these diverse large-scale datasets—from the moderately-sized hERG and Caco-2 to the large-scale AMES and CYP P450 2C19, demonstrates the versatility and scalability of our approach. The consistent improvements across different dataset sizes, ranging from hundreds to over ten thousand molecules, and across both classification and regression tasks, validate that the tri-modal alignment strategy and hierarchical taxonomic annotations provide robust molecular representations that generalize well. These results complement our findings on larger datasets and further establish TRIDENT as a powerful framework for molecular property prediction across the full spectrum of practical applications in drug discovery.

Table 10: Performance on larger-scale TDC datasets. For CYP P450 2C19, we report accuracy (ACC) and ROC-AUC. For Caco-2 regression task, we report RMSE (lower is better).

| Dataset | Metric | MolT5 | MolXPT | BioT5 | BioT5+ | TRIDENT(M-M) |
|---------|--------|-------|--------|-------|--------|--------------|
| CYP P450 2C19 | ACC | 77.86 ± 0.44 | 77.92 ± 0.99 | 76.40 ± 1.14 | 76.43 ± 0.94 | **80.08 ± 0.17** |
|  | ROC-AUC | 87.32 ± 0.34 | 86.87 ± 0.61 | 84.34 ± 0.15 | 84.78 ± 0.32 | **87.50 ± 0.26** |
| Caco-2 | RMSE | **0.41 ± 0.01** | 0.48 ± 0.03 | 0.57 ± 0.03 | 0.60 ± 0.05 | **0.41 ± 0.03** |

Table 11: Ablation study on the impact of LLM-based summarization in HTA generation. Comparison between using raw JSON taxonomic annotations versus LLM-synthesized HTA descriptions across molecular property prediction datasets (ROC-AUC%). Best results in **bold**.

| Method | Input | | | Datasets | | | | |
|--------|-------|------|-----|------|-------|---------|-------|------|
|  | SMILES | Text | HTA | BBBP | Tox21 | ToxCast | Sider | Bace |
| TRIDENT (M-M) w/o LLM | ✓ | ✓ | ✓ | 71.89±0.56 | 79.01±0.33 | 66.86±0.75 | 62.78±0.45 | 81.12±0.69 |
| TRIDENT (M-M) | ✓ | ✓ | ✓ | **73.95±1.01** | **79.36±0.13** | **67.80±0.37** | **63.64±0.56** | **82.39±0.56** |

## E.3 Performance without LLM Summary

To evaluate the contribution of LLM-based summarization in our HTA generation process, we conduct an ablation study comparing the performance of TRIDENT when using raw JSON taxonomic annotations versus LLM-synthesized HTA descriptions. In this experiment, we directly input the structured JSON files containing hierarchical taxonomic paths and descriptions from the 32 classification systems, bypassing the GPT-4o summarization step described in Section 3.1.

The results in Table 11 demonstrate the effectiveness of LLM-based synthesis in our HTA generation pipeline. When using raw JSON taxonomic annotations without LLM summarization (TRIDENT w/o LLM), the model achieves competitive performance but consistently underperforms compared to the full TRIDENT framework across all datasets.

This performance gap highlights several key advantages of LLM-based summarization: (1) **Information Integration**: The LLM synthesis process effectively combines information from multiple taxonomic systems into coherent, contextually rich descriptions that capture cross-domain knowledge spanning chemistry, biology, and pharmacology. (2) **Semantic Coherence**: Raw JSON annotations often contain fragmented or inconsistent terminology across different classification systems, while LLM synthesis produces semantically coherent descriptions that are more amenable to natural language processing. (3) **Contextual Enrichment**: The synthesis process adds relevant contextual information and relationships between different taxonomic levels that may not be explicitly present in individual classification paths.

While the raw taxonomic annotations still provide valuable structural information that outperforms traditional text-only approaches, the LLM synthesis step proves crucial for maximizing the utility of hierarchical taxonomic knowledge in molecular representation learning. This finding validates our design choice to incorporate GPT-4o in the HTA generation pipeline and demonstrates that the additional computational cost of LLM synthesis is justified by the consistent performance improvements across all molecular property prediction tasks.

## E.4 Impact of Tri-modal vs. Concatenated Text Architecture

To validate the necessity of our tri-modal architecture, we conduct an ablation study comparing our approach with a simpler alternative that concatenates HTA and traditional text descriptions into a single textual input. This experiment evaluates whether treating HTA and text as separate modalities provides advantages over a straightforward concatenation approach.

The results in Table 12 demonstrate the effectiveness of our tri-modal architecture over the concatenation approach. The concatenated version (TRIDENT Concatenated) combines HTA and traditional molecular descriptions into a single text input using simple string concatenation with separator tokens, then processes this unified text through the same text encoder used in our tri-modal framework. While this approach still benefits from the rich semantic information in HTA, it consistently underperforms

Table 12: Ablation study comparing tri-modal architecture (SMILES + Text + HTA as separate modalities) versus concatenated text approach (SMILES + concatenated HTA⊕Text as single text modality). The concatenated approach combines HTA and traditional molecular descriptions using string concatenation with separator tokens, while the tri-modal approach processes each information source through separate encoders with volume-based alignment. Performance reported across molecular property prediction datasets using ROC-AUC(%). Best results in **bold**.

| Method | Architecture | | Datasets | | | | |
|---|---|---|---|---|---|---|---|
| | Modalities | Text Processing | BBBP | Tox21 | ToxCast | Sider | Bace |
| TRIDENT (Concatenated) | SMILES + Text | HTA⊕Text | 70.918±0.82 | 76.67±0.59 | 64.59±0.72 | 61.74±0.83 | 79.15±0.69 |
| TRIDENT (M-M) | SMILES + Text + HTA | Separate | **73.95±1.01** | **79.36±0.13** | **67.80±0.37** | **63.64±0.56** | **82.39±0.56** |

the tri-modal architecture across all datasets. These consistent improvements highlight several key advantages of treating HTA and text as separate modalities:

**Modality-Specific Representation Learning**: The tri-modal architecture allows the model to learn distinct representation spaces for hierarchical taxonomic information and functional descriptions. This separation enables the capture of different semantic aspects—taxonomic relationships in HTA versus direct functional properties in traditional text—that may require different representational strategies.

**Enhanced Alignment Flexibility**: The volume-based tri-modal alignment objective can capture complex geometric relationships between SMILES, text, and HTA that are not accessible when HTA and text are merged into a single modality. This geometric awareness enables more nuanced understanding of how molecular structure relates to both functional properties and taxonomic classifications.

**Reduced Information Interference**: Concatenation may lead to interference between the structured, multi-level taxonomic information and the more direct functional descriptions, potentially diluting the distinct contributions of each information source. Separate processing preserves the unique characteristics of each modality.

**Dynamic Weighting Capabilities**: The tri-modal framework allows for dynamic balancing of different information sources during training through our momentum-based mechanism, whereas concatenation fixes the relative importance of HTA and text information at the input level.

These findings validate our design choice to maintain HTA and traditional text as separate modalities, demonstrating that the additional architectural complexity of tri-modal learning is justified by consistent performance gains across all molecular property prediction tasks.

