# OpenReview forum: "TRIDENT: Tri-Modal Molecular Representation Learning with Taxonomic Annotations and Local Correspondence"
_NeurIPS.cc/2025/Conference — NeurIPS 2025 spotlight_

### Official Review · Reviewer_C5UF · 2025-07-02

**Clarity:** 3
**Significance:** 3
**Originality:** 3
**Rating:** 5
**Confidence:** 4

**Summary:**

This study argues that previous models overlook fine-grained annotations and suffer from alignment limitations among modalities, as well as ignoring local fine-grained relationships. To address this issue, this work combines SMILES, textual descriptions, and annotations together to create a high-quality dataset with 47,269 molecules from PubChem. It trains the TRIDENT model using a designed volume-based contrastive loss. Moreover, a novel local alignment module is proposed to link substructures to descriptions. The model achieves SOTA performance across 11 molecular property prediction tasks from TDC and MoleculeNet.

**Questions:**

1. The ablation study evaluates the contribution of each module, but why not provide sensitivity analysis on different hyperparameter settings, such as learning rate or temperature for contrastive loss?
2. Why not compare with stronger and more recent baselines such as Uni-Mol or HiGNN, which are commonly used in recent molecular representation works?
3. This paper mentions multimodal fusion, but why not include comparisons with other recent multimodal models like 3D-MolT5, GIMLET, or MolFM, which are also designed for aligning structure and language?

**Ethical Concerns:**

["NO or VERY MINOR ethics concerns only"]

**Final Justification:**

The authors have addressed my comments so I increased the score to 5.

**Limitations:**

1. The model lacks sensitivity analysis on hyperparameter settings, and we cannot evaluate its robustness or stability under different training conditions.
2. The baselines used in experiments are not the most recent or strongest, and the improvements on most datasets are small, which makes it hard to claim SOTA performance.
3. There is no comparison with other recent multimodal fusion models such as 3D-MolT5, GIMLET, or MolFM, so the advantage of this method in multimodal fusion is not clearly demonstrated.

**Quality:**

3

**Strengths And Weaknesses:**

Strength
1. This work introduces annotation data into the training process, which improves the model’s understanding of molecules and substructures. This is a very promising and innovative direction.
2. To verify the effectiveness of the TRIDENT method, this paper evaluates on 11 datasets from two benchmarks and compares with multiple baselines using sequence inputs, demonstrating the performance of the proposed model.

Weakness:
1. The ablation experiments evaluate the contribution of each module to the overall model and highlight the role of HTA. However, there is no sensitivity analysis regarding model-related settings, such as different hyperparameter choices.
2. In the experimental part, the paper does not compare with newer and stronger baselines such as Uni-Mol and HiGNN. Moreover, the improvements on most datasets are small, which makes it difficult to prove that the model achieves over SOTA performance significantly. It also does not compare with recent multimodal fusion models such as 3D-MolT5 and GIMLET, or MolFM which is mentioned in the related work. The advantage of this paper’s multimodal fusion is not clearly demonstrated.

---

> ### Author Rebuttal · Authors · 2025-07-30
>
> We thank the reviewer for their feedback and highlighting the innovative model design and SOTA performance of our TRIDENT model. We address the valuable questions and comments raised by the reviewer point-by-point:
>
> **1. Sensitivity analysis on hyperparameter settings:** Following the reviewer’s suggestion, we performed sensitivity analysis on key hyperparameters: learning rate, contrastive loss temperature, and the momentum parameter β (used for weighting global and local alignment losses). We conducted this analysis on three MoleculeNet datasets and one TDC dataset (Caco-2). The results show that within a reasonable range of hyperparameters (learning rate 1e−4 to 1e−5, β between 0.5–0.9, and temperature 0.07–0.03), TRIDENT consistently achieves similar performance. This indicates that TRIDENT is robust to hyperparameter choices and does not require tuning for each individual dataset.
> | lr, beta, temperature | Metric | ESOL | FreeSolv | Lipophilicity | Caco-2 |
> |----------------------|--------|------|----------|---------------|--------|
> | 1e-5 & 0.9 & 0.07 (Ours) | RMSE | 0.718±0.072 | 1.424±0.031 | 0.603±0.009 | 0.413±0.03 |
> | 1e-5 & 0.9 & 0.03 | RMSE | 0.723±0.035 | 1.485±0.052 | 0.617±0.004 | 0.419±0.07 |
> | 1e-5 & 0.5 & 0.07 | RMSE | 0.739±0.024 | 1.478±0.076 | 0.619±0.008 | 0.433±0.04 |
> | 1e-4 & 0.5 & 0.07 | RMSE | 0.743±0.040 | 1.482±0.035 | 0.623±0.008 | 0.431±0.05 |
> | 1e-4 & 0.9 & 0.07 | RMSE | 0.731±0.004 | 1.443±0.043 | 0.602±0.005 | 0.418±0.02 |
>
>
> **2. Additional recent or multimodal baselines:** In response to the reviewer’s comment (and also reviewer g9iG), we have extended our comparisons to include recent state-of-the-art baselines, namely Uni-Mol (3D molecular modeling) and multimodal large-scale models such as BioT5, BioT5+, and MolXPT. (The suggested comparison with MolFM is already reported in Table 1 of the paper.) We have now evaluated these new models on MoleculeNet regression tasks and seven TDC datasets, where TRIDENT consistently outperforms these strong baselines. These additional results and corresponding citations to these baselines will be incorporated into the camera-ready version of the paper. For completeness, we also report the results below.
>
>
> *MoleculeNet regression tasks:*
> | Method | ESOL | FreeSolv | Lipophilicity |
> |--------|------|----------|---------------|
> | GCN | 1.43 ± 0.05 | 2.87 ± 0.14 | 0.85 ± 0.08 |
> | GIN | 1.45 ± 0.02 | 2.76 ± 0.18 | 0.85 ± 0.07 |
> | GINE | 0.98 ± 0.10 | 2.92 ± 0.31 | 0.81 ± 0.01 |
> | SR-GINE | 0.73 ± 0.17 | 2.52 ± 0.45 | 0.76 ± 0.02 |
> | Uni-Mol | 0.79 ± 0.03 | 1.48 ± 0.05 | **0.60 ± 0.02** |
> | BioT5 | 0.80 ± 0.02 | 1.63 ± 0.02 | 0.74 ± 0.07 |
> | BioT5+ | 0.79 ± 0.01 | 1.98 ± 0.13 | 0.74 ± 0.06 |
> | MolXPT | 0.75 ± 0.01 | 1.60 ± 0.03 | 0.69 ± 0.01 |
> | MolFormer | 0.78 ± 0.12 | 1.67 ± 0.06 | 0.63 ± 0.02 |
> | MolT5 | 0.82 ± 0.02 | 1.55 ± 0.14 | 0.65 ± 0.04 |
> | TRIDENT (M-M) | **0.72 ± 0.07** | **1.42 ± 0.03** | **0.60 ± 0.01** |
>
>
> *7 TDC datasets:*
> | Dataset | Metric | MolT5 | MolXPT | BioT5 | BioT5+ | TRIDENT (M-M) |
> |---------|--------|-------|--------|-------|--------|---------------|
> | DILI (475 drugs) | ROC-AUC | 77.37±1.15 | 91.67±0.76 | 82.45±1.81 | 82.58±1.65 | **94.56±0.88** |
> | | ACC | 69.44±1.20 | 84.03±3.19 | 76.39±3.18 | 80.56±1.20 | **86.80±3.18** |
> | Carcinogens (278 drugs) | ROC-AUC | 86.89±1.00 | 75.76±2.73 | 82.83±4.31 | 86.62±2.32 | **87.07±0.77** |
> | | ACC | 84.45±1.11 | 80.90±2.06 | 76.19±2.06 | 77.41±2.00 | **84.62±1.07** |
> | Skin Reaction (404 drugs) | ROC-AUC | 68.67±3.99 | 61.08±1.28 | 68.27±4.41 | 65.25±0.66 | **72.00±1.09** |
> | | ACC | 62.22±1.41 | **62.60±1.40** | 62.21±1.06 | 62.27±1.20 | **62.60±1.40** |
> | hERG (648 drugs) | ROC-AUC | 76.25±1.22 | 82.44±2.14 | 77.48±1.59 | 82.27±3.29 | **83.31±1.63** |
> | | ACC | 77.04±4.90 | 81.31±2.31 | 80.30±3.03 | 80.31±1.52 | **83.33±2.62** |
> | AMES (7,255 drugs) | ROC-AUC | 76.93±0.84 | 83.73±0.69 | 77.57±0.69 | 78.18±1.48 | **86.87±0.60** |
> | | ACC | 70.87±2.22 | 76.50±2.06 | 73.25±1.67 | 73.30±1.15 | **80.20±1.44** |
> | CYP P450 2C19 (12,665 drugs) | ROC-AUC | 87.32±0.34 | 86.87±0.61 | 84.34±0.15 | 84.78±0.32 | **87.50±0.26** |
> | | ACC | 77.86±0.44 | 77.92±0.99 | 76.40±1.14 | 76.43±0.94 | **80.08±0.17** |
> | Caco-2 (906 drugs) | RMSE $\downarrow$ | **0.41±0.01** | 0.48±0.03 | 0.57±0.03 | 0.60±0.05 | **0.41±0.03** |
>
> We thank the reviewer again for their questions! Given that we have addressed all your questions, we request you to kindly consider raising your score!

---

> > ### Comment · Reviewer_C5UF · 2025-08-03
> > **Thanks for the additional experiments**
> >
> > I thank the authors for providing the new experimental results. Although the proposed method demonstrates some performance improvement, the gains appear marginal compared to the stronger baselines. In particular, it is unclear why the authors did not include UniMol, which looks to be the best baseline in the authors' MoleculeNet dataset results, when they analyzed the TDC dataset. Additionally, prior literature has highlighted numerous limitations and issues associated with the MoleculeNet dataset, which further underscores the importance of thoroughly evaluating the proposed approach against robust and relevant benchmarks.

---

> ### Author Response · Authors · 2025-08-04
> **Authors' response to additional experiments (Comparison with Uni-Mol baseline on TDC)**
>
> We thank the reviewer for the follow-up comment.
>
> **Comparison against Uni-Mol baseline on TDC datasets:** Due to the limited rebuttal period, we initially prioritized experiments that addressed multiple reviewer comments simultaneously, which led us to focus on Uni-Mol primarily for the MoleculeNet benchmarks. Following the reviewer’s suggestion, we have now run Uni-Mol on the TDC benchmarks and summarize the results below:
>
> | Dataset | Metric | Uni-Mol | TRIDENT (M-M) | Improvement% |
> |---------|--------|--------|---------------|-------|
> | DILI (475 drugs) | ROC-AUC | 93.63±1.05 | **94.56±0.88** | +0.99% |
> |                 | ACC     | 81.25±3.61 | **86.80±3.18** | +6.83% |
> | Carcinogens (278 drugs) | ROC-AUC | 77.02±2.32 | **87.07±0.77** | +13.05% |
> |                        | ACC     | 81.11±1.05 | **84.62±1.07** | +4.33% |
> | Skin Reaction (404 drugs) | ROC-AUC | 71.17±1.91 | **72.00±1.09** | +1.17% |
> |                          | ACC     | 62.37±1.03 | **62.60±1.40** | +0.37% |
> | hERG (648 drugs) | ROC-AUC | 79.31±2.74 | **83.31±1.63** | +5.04% |
> |                 | ACC     | 82.32±0.87 | **83.33±2.62** | +1.23% |
> | AMES (7,255 drugs) | ROC-AUC | 79.43±0.92 | **86.87±0.60** | +9.37% |
> |                   | ACC     | 72.57±1.10 | **80.20±1.44** | +10.51% |
> | CYP P450 2C19 (12,665 drugs) | ROC-AUC | 83.59±0.74 | **87.50±0.26** | +4.68% |
> |                           | ACC     | 76.87±1.71 | **80.08±0.17** | +4.18% |
> | Caco-2 (906 drugs) | RMSE ↓ | 0.53±0.01 | **0.41±0.03** | +22.64% |
>
> These results show **consistent improvements across all seven TDC benchmarks, with gains as high as 22.64% (Caco-2) and 13.05% (Carcinogens)**.
>
> We also agree that MoleculeNet has known limitations, which is why we conducted comprehensive experiments on seven TDC benchmarks additionally. We hope this addresses the reviewer’s questions and concerns fully. If so, we would kindly appreciate reconsideration of the overall score!

---

> > ### Comment · Reviewer_C5UF · 2025-08-04
> > **Thanks.**
> >
> > I have increased my score to 5.

---

> > > ### Author Response · Authors · 2025-08-05
> > > **Thank you for increasing your score!**
> > >
> > > We appreciate the reviewer’s thoughtful comments and are glad that our responses helped clarify the concerns. Thank you for the positive evaluation.

---

### Official Review · Reviewer_g9iG · 2025-07-02

**Clarity:** 3
**Significance:** 3
**Originality:** 3
**Rating:** 5
**Confidence:** 4

**Summary:**

This paper introduce a molecular representation learning model that utilize three types of data, SMILES, text and hierarchical taxonomic annotation (HTA). HTA is a new datatype that is used in molecular representation learning, and the author curated a dataset with HTA as well as SMILES and text of 47,269 molecules. A new contrastive learning loss for multimodal alignment, Global Volume-based Contrastive Loss, is utilized to train train the model with three data types. In addition, fine-grained local alignment between molecular substructures and the relevant text description is explored in model training.  Experiments are conducted on two public datasets for molecular property learning and the proposed method outperform a series of baseline methods.

**Questions:**

please respond to the weakness points

**Ethical Concerns:**

["NO or VERY MINOR ethics concerns only"]

**Final Justification:**

I raise my score from 4 to 5 after the response with additional latest baseline methods, experiments on new tasks and more ablation studies.

**Limitations:**

Yes

**Paper Formatting Concerns:**

The Appendix is put in a seperate file.

**Quality:**

3

**Strengths And Weaknesses:**

Strength
1.The curated dataset with many different hierarchical taxonomic annotation are valuable for the field.

2.The motivation of using multimodal alignment to train a molecular representation learning model with three modality is reasonable.

3.The idea of align molecular substructures with their function description is valuable.

4.The performance of the proposed method is superior over baseline methods with the same available data.



Weakness.
1.With the MoleculeNet dataset, there are a series of regression tasks, which are also used for evaluating molecular property prediction models. But this proposed method is not evaluated on these tasks.

2.There are approximately 380k SMILES-text pairs in the original dataset, from witch the authors curated 47,269 SMILES-Text-HTA triplets. The size of the triplet dataset is much smaller than the original dataset, which is a negative factor for representation learning. Two other kinds of ablation should be tried. First, adjust the proposed architecture to exclude HTA while utilize the whole data of 380k SMILES-text pairs. Second, on the base of the current proposed model, besides using the data of SMILES-Text-HTA triplets, adding different numbers of SMILES-text pairs data for model training.

3.The overall performance of the proposed method is superior than the baselines with the same data available. However, there are some other studies using 3D molecular graph as input, which may also achieve high performance. This line of research should be discussed. A lower performance than the competing research line will also reduce the practical value of the proposed method.

4. Comparison to more recent studies is needed. Such as the methods in the following works.
[1]Pei Q, Wu L, Gao K, et al. BioT5+: Towards Generalized Biological Understanding with IUPAC Integration and Multi-task Tuning[C]//ACL (Findings). 2024.
[2]Pei Q, Zhang W, Zhu J, et al. BioT5: Enriching Cross-modal Integration in Biology with Chemical Knowledge and Natural Language Associations[C]//The 2023 Conference on Empirical Methods in Natural Language Processing.
[3]Liu Z, Zhang W, Xia Y, et al. MolXPT: Wrapping Molecules with Text for Generative Pre-training[C]//The 61st Annual Meeting Of The Association For Computational Linguistics. 2023.

---

> ### Author Rebuttal · Authors · 2025-07-30
>
> We thank the reviewer for their positive feedback and highlighting the value of our hierarchical taxonomic annotations for the field as well as for our substructure-subtext alignment approach which obtains SOTA performance on molecular property prediction benchmarks. We address the valuable questions and comments raised by the reviewer grouping similar questions together:
>
> **1. Results on MoleculeNet regression tasks and discussion/comparison with 3D models:** In addition to the MoleculeNet classification tasks reported in the main text, we have now also evaluated TRIDENT on three MoleculeNet regression tasks, and we report the results below (metric is RMSE). For a more comprehensive comparison, we extend the baseline set beyond those in the main text to include: (a) 2D topological graph models (GCN, GIN, GINE, SR-GINE) and 3D molecular graph models (Uni-Mol). These additional results demonstrate TRIDENT’s strong performance over existing and recent baselines (including recent 2D and 3D graph models ). We will cite and include the discussion of these models in the camera-ready version along with the new results on MoleculeNet regression tasks shown here.
> | Method | ESOL | FreeSolv | Lipophilicity |
> |--------|------|----------|---------------|
> | GCN | 1.43 ± 0.05 | 2.87 ± 0.14 | 0.85 ± 0.08 |
> | GIN | 1.45 ± 0.02 | 2.76 ± 0.18 | 0.85 ± 0.07 |
> | GINE | 0.98 ± 0.10 | 2.92 ± 0.31 | 0.81 ± 0.01 |
> | SR-GINE | 0.73 ± 0.17 | 2.52 ± 0.45 | 0.76 ± 0.02 |
> | Uni-Mol | 0.79 ± 0.03 | 1.48 ± 0.05 | **0.60 ± 0.02** |
> | BioT5 | 0.80 ± 0.02 | 1.63 ± 0.02 | 0.74 ± 0.07 |
> | BioT5+ | 0.79 ± 0.01 | 1.98 ± 0.13 | 0.74 ± 0.06 |
> | MolXPT | 0.75 ± 0.01 | 1.60 ± 0.03 | 0.69 ± 0.01 |
> | MolFormer | 0.78 ± 0.12 | 1.67 ± 0.06 | 0.63 ± 0.02 |
> | MolT5 | 0.82 ± 0.02 | 1.55 ± 0.14 | 0.65 ± 0.04 |
> | TRIDENT (M-M) | **0.72 ± 0.07** | **1.42 ± 0.03** | **0.60 ± 0.01** |
>
> **2. Comparison with BioT5, BioT5+, MolXPT:** We also compared TRIDENT against models BioT5, BioT5+, and MolXPT as suggested by the reviewer. We reported the comparison in the table above for three MoleculeNet regression tasks. We  are also including the results for 7 TDC datasets in the table below. TRIDENT outperforms the baselines here as well. We will also cite BioT5, BioT5+, and MolXPT in the camera-ready version and will include these results.
> | Dataset | Metric | MolT5 | MolXPT | BioT5 | BioT5+ | TRIDENT (M-M) |
> |---------|--------|-------|--------|-------|--------|---------------|
> | DILI (475 drugs) | ROC-AUC | 77.37±1.15 | 91.67±0.76 | 82.45±1.81 | 82.58±1.65 | **94.56±0.88** |
> | | ACC | 69.44±1.20 | 84.03±3.19 | 76.39±3.18 | 80.56±1.20 | **86.80±3.18** |
> | Carcinogens (278 drugs) | ROC-AUC | 86.89±1.00 | 75.76±2.73 | 82.83±4.31 | 86.62±2.32 | **87.07±0.77** |
> | | ACC | 84.45±1.11 | 80.90±2.06 | 76.19±2.06 | 77.41±2.00 | **84.62±1.07** |
> | Skin Reaction (404 drugs) | ROC-AUC | 68.67±3.99 | 61.08±1.28 | 68.27±4.41 | 65.25±0.66 | **72.00±1.09** |
> | | ACC | 62.22±1.41 | **62.60±1.40** | 62.21±1.06 | 62.27±1.20 | **62.60±1.40** |
> | hERG (648 drugs) | ROC-AUC | 76.25±1.22 | 82.44±2.14 | 77.48±1.59 | 82.27±3.29 | **83.31±1.63** |
> | | ACC | 77.04±4.90 | 81.31±2.31 | 80.30±3.03 | 80.31±1.52 | **83.33±2.62** |
> | AMES (7,255 drugs) | ROC-AUC | 76.93±0.84 | 83.73±0.69 | 77.57±0.69 | 78.18±1.48 | **86.87±0.60** |
> | | ACC | 70.87±2.22 | 76.50±2.06 | 73.25±1.67 | 73.30±1.15 | **80.20±1.44** |
> | CYP P450 2C19 (12,665 drugs) | ROC-AUC | 87.32±0.34 | 86.87±0.61 | 84.34±0.15 | 84.78±0.32 | **87.50±0.26** |
> | | ACC | 77.86±0.44 | 77.92±0.99 | 76.40±1.14 | 76.43±0.94 | **80.08±0.17** |
> | Caco-2 (906 drugs) | RMSE $\downarrow$ | **0.41±0.01** | 0.48±0.03 | 0.57±0.03 | 0.60±0.05 | **0.41±0.03** |
>
> **3. Ablations (a) with 380k SMILES-text original dataset, (b) using further fine-tuning with larger SMILES-text dataset starting with pre-trained TRIDENT weights:** We appreciate the reviewer’s suggestion to explore this. For this ablation, we pre-trained TRIDENT on the full 380k SMILES–Text pairs without HTA and evaluated it on three MoleculeNet regression tasks and one TDC dataset (Caco-2). Training solely on the full 380k SMILES–Text pairs degrades performance due to known dataset issues (artificially templated text, limited semantic richness (too short descriptions), and duplicates) [1]. For the second ablation, we used the model weights from triplet-based trained TRIDENT model and fine-tuned it with remaining (380k-47,269) SMILES-text pairs. The performance improved slightly but still lagged behind TRIDENT trained solely on curated triplets as fine-tuning on potentially low quality data is detrimental for learning. This confirms the value of our data curation contribution and the inclusion of hierarchical taxonomic annotations (HTA) for representation learning.
> | Method | Metric | ESOL | FreeSolv | Lipophilicity | Caco-2 |
> |--------|--------|------|----------|---------------|--------|
> | 380k SMILES-text | RMSE | 0.78 ± 0.13 | 1.68 ± 0.07 | 0.67 ± 0.05 | 0.53 ± 0.06 |
> | Finetune | RMSE | 0.75 ± 0.06 | 1.61 ± 0.15 | 0.65 ± 0.13 | 0.51 ± 0.05 |
> | Ours | RMSE | **0.72 ± 0.07** | **1.42 ± 0.03** | **0.60 ± 0.01** | **0.41 ± 0.03** |
>
>
> We believe we have addressed all your questions and request the reviewer if they may consider raising their score!
>
> [1]: Ross, Jerret, et al. "Large-scale chemical language representations capture molecular structure and properties." Nature Machine Intelligence 4.12 (2022): 1256-1264.

---

> > ### Comment · Reviewer_g9iG · 2025-08-07
> > **comments to response**
> >
> > I appriciate the new experimental results the authors added and my concerns have been well addressed. By also considering the comments from other reviewers, I'd like to raise my final score.

---

> ### Author Response · Authors · 2025-08-07
> **Friendly request for feedback on responses to your questions...**
>
> Dear reviewer,
>
> We greatly appreciate the time and effort you’ve put into reviewing our submission. As the discussion period is nearing conclusion, we kindly request your feedback regarding our detailed responses to your valuable comments. We believe we have effectively addressed all questions. Please let us know if you have any further questions. We’d be happy to clarify. Otherwise, if your concerns have been addressed, we’d greatly appreciate it if you could consider raising the score. Thank you!

---

### Official Review · Reviewer_tPkn · 2025-07-02

**Clarity:** 3
**Significance:** 2
**Originality:** 3
**Rating:** 5
**Confidence:** 3

**Summary:**

The paper presents TRIDENT, a novel framework for molecular property prediction that integrates three modalities: molecular SMILES, textual descriptions, and hierarchical taxonomic annotations (HTA). The authors introduce a geometry-aware volume-based contrastive loss for global alignment and a local alignment module that links molecular substructures to corresponding textual descriptions.

**Questions:**

- There are so many datasets in TDC; why did you choose these three?

- A considerable portion of the paper is dedicated to explaining global tri-modal alignment, but there is less discussion on local alignment. Is there a detailed ablation study on local alignment?

**Ethical Concerns:**

["NO or VERY MINOR ethics concerns only"]

**Final Justification:**

The construction and application of Hierarchical Taxonomic Annotations (HTA) are straightforward and have been shown to be useful across multiple datasets. The other reviewers' comments are generally positive as well, and no significant flaws have been identified.

**Limitations:**

see weakness

**Quality:**

3

**Strengths And Weaknesses:**

Strengths

- TRIDENT's tri-modal approach is novel, combining SMILES, textual descriptions, and HTA for richer molecular representations. The use of HTA adds a structured, multi-level understanding that previous methods lacked.

- The model outperforms several strong baselines on multiple molecular property prediction tasks.

- The introduction of both global and local alignment strategies, balanced via a momentum-based mechanism, addresses the alignment challenges across different modalities effectively.

Weakness

- While the framework shows impressive results on smaller datasets, there is limited discussion on the scalability of the approach when applied to much larger molecular datasets and other downstream tasks.

---

> ### Author Rebuttal · Authors · 2025-07-30
>
> We thank the reviewer for appreciating the novelty and highlighting the state-of-the-art performance of our proposed model on molecular property prediction tasks. We also thank the reviewer for raising valuable questions. We address all questions raised by the reviewer point-by-point.
>
> **1. Why chose only 3 TDC datasets/more TDC datasets/other downstream tasks on larger datasets:** We actually evaluated TRIDENT on five TDC datasets: three in the main text (as noted by the reviewer) and two additional ones in Appendix Table 8. The appendix includes results on larger datasets (AMES with 7,255 drugs and hERG with 648 drugs), where TRIDENT also outperforms all baselines. Additionally, for the rebuttal, we conducted experiments on two other larger datasets (CYP P450 2C19 with 12,665 drugs and Caco-2 with 906 drugs ), bringing the total to seven TDC datasets. For completeness, we report all results below, which further confirm TRIDENT’s consistent performance across diverse TDC benchmarks.
> | Dataset | Metric | MolT5 | MolXPT | BioT5 | BioT5+ | TRIDENT (M-M) |
> |---------|--------|-------|--------|-------|--------|---------------|
> | DILI (475 drugs) | ROC-AUC | 77.37±1.15 | 91.67±0.76 | 82.45±1.81 | 82.58±1.65 | **94.56±0.88** |
> | | ACC | 69.44±1.20 | 84.03±3.19 | 76.39±3.18 | 80.56±1.20 | **86.80±3.18** |
> | Carcinogens (278 drugs) | ROC-AUC | 86.89±1.00 | 75.76±2.73 | 82.83±4.31 | 86.62±2.32 | **87.07±0.77** |
> | | ACC | 84.45±1.11 | 80.90±2.06 | 76.19±2.06 | 77.41±2.00 | **84.62±1.07** |
> | Skin Reaction (404 drugs) | ROC-AUC | 68.67±3.99 | 61.08±1.28 | 68.27±4.41 | 65.25±0.66 | **72.00±1.09** |
> | | ACC | 62.22±1.41 | **62.60±1.40** | 62.21±1.06 | 62.27±1.20 | **62.60±1.40** |
> | hERG (648 drugs) | ROC-AUC | 76.25±1.22 | 82.44±2.14 | 77.48±1.59 | 82.27±3.29 | **83.31±1.63** |
> | | ACC | 77.04±4.90 | 81.31±2.31 | 80.30±3.03 | 80.31±1.52 | **83.33±2.62** |
> | AMES (7,255 drugs) | ROC-AUC | 76.93±0.84 | 83.73±0.69 | 77.57±0.69 | 78.18±1.48 | **86.87±0.60** |
> | | ACC | 70.87±2.22 | 76.50±2.06 | 73.25±1.67 | 73.30±1.15 | **80.20±1.44** |
> | CYP P450 2C19 (12,665 drugs) | ROC-AUC | 87.32±0.34 | 86.87±0.61 | 84.34±0.15 | 84.78±0.32 | **87.50±0.26** |
> | | ACC | 77.86±0.44 | 77.92±0.99 | 76.40±1.14 | 76.43±0.94 | **80.08±0.17** |
> | Caco-2 (906 drugs) | RMSE $\downarrow$ | **0.41±0.01** | 0.48±0.03 | 0.57±0.03 | 0.60±0.05 | **0.41±0.03** |
>
> In case the reviewer referred to large molecules such as proteins by larger datasets, we emphasize that since the pre-training is done on small molecules data curated from PubChem, we have benchmarked the model on many tasks from commonly used small molecule property prediction benchmarks (MoleculeNet and TDC) relevant for small molecule drug-discovery applications. Extending TRIDENT to larger molecules (such as proteins) is an interesting future work!
>
> **2. Ablation and discussion on local alignment hyperparameters:** The local alignment module has only two main hyperparameters:(a) the relative weighting of local vs. global alignment terms, (b) the aggregation (pooling) strategy for consolidating sub-text and sub-structure representations.
>
>   a. In the main text (Table 4), we already presented ablation results for different weighting strategies for local vs. global alignment terms, showing that momentum-based dynamic weighting performs best.
>
>   b. Following the reviewer’s suggestion, we additionally ablated the aggregation strategy, comparing multiple pooling operators. Note that the “weighted mean pooling” variant refers to assigning weights based on the frequency of functional groups present in the molecule. The results (reported below: three MoleculeNet regression tasks and one TDC dataset (Caco-2)) indicate that TRIDENT’s performance is similar irrespective of the choice of aggregation function.
> | Method | Metric | ESOL | FreeSolv | Lipophilicity | Caco-2 |
> |--------|--------|------|----------|---------------|--------|
> | Max pooling | RMSE | 0.72 ± 0.07 | 1.42 ± 0.03 | 0.60 ± 0.01 | 0.41 ± 0.03 |
> | Mean pooling | RMSE | 0.73 ± 0.02 | 1.44 ± 0.02 | 0.62 ± 0.02 | 0.42 ± 0.04 |
> | Weighted mean pooling | RMSE | 0.71 ± 0.04 | 1.42 ± 0.03 | 0.61 ± 0.01 | 0.40 ± 0.02 |
>
> We will include these results in the Appendix of camera-ready version. Given that we have addressed all your questions, we request if the reviewer may kindly consider raising their score!

---

> > ### Comment · Reviewer_tPkn · 2025-08-03
> >
> > Thank you for the detailed explanation. The result is convincing, and I will increase my score.

---

> > > ### Author Response · Authors · 2025-08-04
> > > **Thank you for increasing your score!**
> > >
> > > We thank the reviewer for increasing their score and we are happy to have addressed their questions in full. Thank you for your thoughtful review and suggestions!

---

### Official Review · Reviewer_TSPM · 2025-07-20

**Clarity:** 4
**Significance:** 4
**Originality:** 4
**Rating:** 6
**Confidence:** 3

**Summary:**

TRIDENT introduces a novel trimodal molecular representation learning framework that integrates SMILES structures, textual descriptions and Hierarchical Taxonomic Annotations (HTA) to overcome limitations in existing methods. The framework employs a geometry aware volume based contrastive loss for global trimodal alignment and a local alignment module linking RDKit identified substructures to text spans, dynamically balanced via a momentum mechanism. The paper achieves state-of-the-art performance across 11 molecular property prediction benchmarks from MoleculeNet, establishing a new paradigm for structured and biologically meaningful molecular embeddings while releasing a comprehensive taxonomy annotated dataset for community use.

**Questions:**

- What is the computational overhead introduced by the volume-based contrastive loss and local alignment module? How does training time and memory usage compare to pairwise contrastive models like MoMu or MolFM?
- Could you provide more detailed information about the prompting strategy, filtering criteria, and validation process used to synthesize the HTA texts with GPT-4o? Are the prompts and outputs deterministic or do they vary across runs? How were factual hallucinations and taxonomic inconsistencies addressed?

**Ethical Concerns:**

["NO or VERY MINOR ethics concerns only"]

**Final Justification:**

The paper is well detailed and the authors have well addressed the concerns raised in the initial review.

**Limitations:**

- The paper does not provide sufficient error analysis. Although results are strong, the paper lacks insight into when TRIDENT fails.
- Including generalizability experiments will be helpful. Performance on non-taxonomically rich or non-biomedical datasets would test the universality of the method.
- While the model performs well, there’s no qualitative or visualization-based analysis of the embedding space, cross-modal attention, or interpretability of predictions

**Quality:**

4

**Strengths And Weaknesses:**

Strengths
- The framework unifies SMILES, text and hierarchical taxonomies (HTA) addressing a critical gap in molecular representation. The HTA’s multi-level annotations capture nuanced functional semantics beyond flat descriptions
- The alignment mechanisms are novel. Volume-based contrastive loss enables geometry-aware trimodal alignment outperforming pairwise methods, local substructure-text alignment resolves granular correspondence and the momentum based balancing dynamically optimizes global vs. local objectives.
- Ablation studies are thorough and clearly show performance drop when removing HTA, local alignment, or the volume loss

Weaknesses
- It's unclear how deterministic or reproducible this GPT-based synthesis is. GPT-generated text could introduce hallucinations or inconsistent phrasing, possibly leading to poor generalization or overfitting
- The volume-based contrastive loss and functional group alignment introduce non-trivial computational overhead
- The paper does not clearly explain the architecture of the encoders and projectors (e.g., how SciBERT or MolT5 is fine-tuned or frozen, the tokenization process for HTA).

---

> ### Author Rebuttal · Authors · 2025-07-30
>
> We thank the reviewer for their positive feedback and appreciating the novelty of our method. We address the valuable comments and questions raised by the reviewer point-by-point and will include them at appropriate places in camera-ready version.
>
> **1. Training time and memory usage:** We use pre-trained frozen encoders for SMILES, HTA, and functional descriptions, with only lightweight trainable projection layers (2.5 M parameters vs. 223 M in MoMu). This parameter-efficient design significantly reduces computational requirements for training and deployment. Under the same experimental settings, MoMu processes 32.5 samples per second while our method processes 16.4 samples per second on a single NVIDIA H100 GPU, with the difference primarily attributed to processing additional modalities (functional groups and HTA information). The total GPU memory usage is ~45 GB versus ~35 GB for MoMu, with the modest increase due to loading multiple frozen encoders and handling additional data modalities, but still easily affordable by most researchers.
>
> **2. Details on prompting strategy and validation process to synthesize HTA texts with GPT-4o:**
>
> *Prompting strategy:* The exact prompt template used is shown in Appendix A, Figure 4. This template explicitly instructs the model to summarize taxonomic annotations from PubChem without introducing additional information. All HTA summaries were generated using the same fixed prompt and model parameters (temperature=0.2, frequency_penalty = 0.0).
>
> *Filtering and validation:* Because GPT-4o was summarizing well-defined, scientifically validated PubChem Taxonomic Annotations, the risk of introducing extraneous information was minimized. Additionally, we conducted a validation step where domain experts independently reviewed a random sample of 1,000 HTA summaries, comparing them against the corresponding PubChem annotations. No factual hallucinations or taxonomic inconsistencies were observed in this review.
>
> **3. Error analysis (failure cases):** We thank the reviewer for highlighting this important point. While TRIDENT demonstrates strong overall performance, we note a performance decline on specific downstream tasks. For example, in the SIDER dataset, it achieves a ROC-AUC of 0.5904 on the *Product issues task*, which is notably lower than the 0.7383 ROC-AUC on the *Blood and Lymphatic System Disorders task*. A key factor is class imbalance in the former dataset, for which TRIDENT does not currently employ explicit rebalancing mechanisms. Addressing such imbalance-aware training strategies (e.g., loss reweighting or data augmentation) is a promising direction for future work to combine with TRIDENT.
>
> **4. Analysis of the embedding space:** We thank the reviewer for this suggestion. To analyze the embedding space, we calculated the average Euclidean distance between positive pairs across all modalities during pretraining. The distances between positive pairs across all modality combinations consistently decrease throughout training: SMILES-Text distances reduce from 0.7930 (Epoch 1) to 0.4805 (Epoch 40), SMILES-HTA distances decrease from 0.8269 to 0.4708, and Text-HTA distances drop from 0.9511 to 0.8698, with the overall average distance declining from 0.8570 to 0.6070. This systematic reduction in intra-sample cross-modal distances indicates that TRIDENT successfully learns to align semantically related representations across different modalities. We will include this analysis in camera-ready version. Given the rebuttal policy, we cannot include figures in this response but we will also include tSNE projections of embeddings and crossmodal attention heatmaps in the camera-ready version to further illustrate interpretability.
>
> **5. Performance on non-biomedical or non-taxonomic datasets:** We thank the reviewer for this valuable suggestion. Our work is primarily focused on molecular property prediction for drug discovery applications. Exploring non-biomedical datasets is an interesting future direction but is beyond the current scope. Notably, taxonomy information is used only during pre-training; all downstream molecular property prediction tasks rely solely on the SMILES encoder and projector, demonstrating that TRIDENT performs effectively even when taxonomic information is unavailable at inference time.
>
> **6. Architecture of encoders and projectors:** We thank the reviewer for the suggestion. Our model architecture and training configuration are described in Appendix B.2 and Algorithm 2 (Appendix page 20). Briefly, we use pre-trained MoLFormer for SMILES and MolT5/SciBERT for text (including HTA descriptions), both of which are frozen during training. We train only lightweight modality-specific projection heads (three linear layers with GELU, LayerNorm, and dropout) to map encoder outputs to a shared 512-dimensional space. For HTA and Text tokenization, we adopt the default SciBERT WordPiece vocabulary (scivocab) tokenizer, ensuring consistency with its pre-training. When using MolT5 as the HTA and Text encoder, we employ its pre-trained T5Tokenizer. This design minimizes computational overhead while retaining rich semantic representations.

---

> ### Comment · Reviewer_TSPM · 2025-08-09
>
> Thank you for the explanations and for taking the time to answer. My concerns are well addressed.

---

### Decision · Program_Chairs · 2025-09-17

**Decision:**

Accept (spotlight)

**Comment:**

This paper proposes a molecular representation learning framework, TRIDENT, which leverages three modalities: SMILES (molecular structure), Molecular Functional Description (text), and Hierarchical Taxonomic Annotation (text). To this end, the authors introduce a newly curated, high-quality multi-modal molecular dataset and propose two learning objectives: (1) geometry-based global alignment across the three modalities, and (2) fine-grained local alignment based on functional group structures.

The authors successfully addressed the reviewers' concerns during the discussion phase, and as a result, all reviewers unanimously support acceptance. AC also agrees with the strengths highlighted by the reviewers:
- A novel and effective multi-modal molecular representation learning approach that integrates SMILES, textual descriptions, and HTA for richer representations,
- Strong experimental results across a wide range of benchmarks and recent baselines,
- A newly curated multi-modal dataset that includes informative hierarchical annotations.

Therefore, AC recommends acceptance of this submission. The authors are strongly encouraged to incorporate the reviewers' feedback, especially additional experimental results on new benchmarks and baselines, as doing so would further strengthen and clarify the contributions of this work.